# Planning with Large Language Models for Code Generation

**Shun Zhang, Zhenfang Chen, Yikang Shen**
MIT-IBM Watson AI Lab

**Mingyu Ding**
The University of Hong Kong

**Joshua B. Tenenbaum**
MIT BCS, CBMM, CSAIL

**Chuang Gan**
UMass Amherst, MIT-IBM Watson AI Lab

## Abstract

Existing large language model-based code generation pipelines typically use beam search or sampling algorithms during the decoding process. Although the programs they generate achieve high token-matching-based scores, they often fail to compile or generate incorrect outputs. The main reason is that conventional Transformer decoding algorithms may not be the best choice for code generation. In this work, we propose a novel Transformer decoding algorithm, Planning-Guided Transformer Decoding (PG-TD), that uses a planning algorithm to do lookahead search and guide the Transformer to generate better programs. Specifically, instead of simply optimizing the likelihood of the generated sequences, the Transformer makes use of a planner to generate candidate programs and test them on public test cases. The Transformer can therefore make more informed decisions and generate tokens that will eventually lead to higher-quality programs. We also design a mechanism that shares information between the Transformer and the planner to make our algorithm computationally efficient. We empirically evaluate our framework with several large language models as backbones on public coding challenge benchmarks, showing that 1) it can generate programs that consistently achieve higher performance compared with competing baseline methods; 2) it enables controllable code generation, such as concise codes and highly-commented codes by optimizing modified objective[1].

## 1 Introduction

Large language models like Transformer (Vaswani et al., 2017a) have shown successes in natural language processing, computer vision, and various other domains. Thanks to Transformer's power on sequence modeling, it has been adopted for code generation (Wang et al., 2021; Ahmad et al., 2021) by treating programs as text sequences. Transformer has achieved significant improvements on the benchmarking tasks of code translation (Roziere et al., 2022), code completion (Chen et al., 2021a), and solving coding challenge problems (Hendrycks et al., 2021). Recently, AlphaCode (Li et al., 2022) even achieved a competitive-level performance in programming competitions with the help of large Transformer models pre-trained on a large programming corpus.

Transformer-based pipelines like AlphaCode follow the tradition of natural language processing and use sampling methods (Fan et al., 2018; Dabre & Fujita, 2020) during the generation process. Specifically, they sample a large number of complete programs using a pre-trained code generation Transformer, evaluate these programs using the public test cases provided in the dataset, and output the program that passes the most number of test cases. Compared with beam search-based methods, these sampling followed by filtering algorithms (which we will refer to as *sampling + filtering*) can take advantage of test cases and indeed improve the quality of the generated programs. However, during the Transformer generation process, they do not consider the test cases at all. Instead, they only use the test cases to evaluate the programs after all the candidate programs are generated. This can make their algorithms sample inefficient. Different from natural languages, programs may fail

---

[1]Project page: https://codeaimcts.github.io. Correspondence to: shun.zhang@ibm.com.

completely with even a single incorrect generated token. So these algorithms need to exhaustively sample a large number of programs to find a correct solution.

The main reason behind the sample efficiency issue of these algorithms is that the Transformer beam search algorithm and the sampling algorithm (Vaswani et al., 2017b) may not be the best choices for code generation. An ideal code generation algorithm should stop early in the generation process when it knows the program it currently generates would certainly fail, and bias the generation process towards generating successful programs that pass more test cases. To achieve such a goal, we contribute to applying a planning algorithm in the Transformer generation process. Since a planning algorithm can use the pass rates of the generated programs as its objective, we use it to determine the quality of the generated codes and make the Transformer model make more informed decisions.

In this paper, we investigate the following research question: *Can we integrate a planning algorithm with a pre-trained code generation Transformer, achieving an algorithm that generates better programs than the conventional Transformer generation algorithms and the well-accepted sampling + filtering scheme in the literature?* To answer this question, we propose a novel algorithm, *Planning-Guided Transformer Decoding* (PG-TD). During the code generation process, a planner does lookahead search and finds tokens that will lead to high-quality codes. The planner alone may not efficiently find high-quality codes due to the large search space of codes, and that is where a pre-trained code generation Transformer comes into play. Specifically, the Transformer beam search algorithm and the next-token probabilities are used inside the planner to provide useful heuristics. We find that a straightforward integration between the planner and the Transformer can be computationally inefficient. So we design mechanisms that allow the Transformer and the planner to share their information to make the overall algorithm more efficient.

We emphasize that our algorithm is *model-agnostic*, that is, any standard code generation Transformer model can be used as the backbone Transformer. Importantly, our algorithm does not require acquiring more sample solutions or finetuning the Transformer model to improve its performance. We empirically find that our proposed algorithm generates higher-quality programs under multiple accepted metrics compared with competing baseline methods. Additionally, we also empirically show that our algorithm has the following advantages. 1) By changing the reward function of the planner, our algorithm becomes versatile and can optimize different objective functions without the necessity of finetuning the Transformer model. 2) Our algorithm can generate solutions that are used to finetune a code-generation Transformer model to improve the Transformer's performance. More precisely, we have the following contributions in this paper.

- First, we propose a novel algorithm, Planning-Guided Transformer Decoding (PG-TD), that uses a planning algorithm for lookahead search and guide the Transformer to generate better codes. Our algorithm is model-agnostic, which can work with any standard Transformer model, and does not require knowledge of the grammar of the generated programs.
- Second, a direct integration of the planning algorithm with the Transformer decoding process can cause redundant uses of the Transformer beam search algorithm. We contribute to designing mechanisms that significantly improve the computational efficiency of the algorithm.
- Third, we evaluate our algorithm on competitive programming benchmarks and empirically show that our algorithm can consistently generate better programs in terms of the pass rate and other metrics compared with the baseline methods. We also show that our algorithm is versatile and can optimize objectives other than the pass rate for controllable code generation, such as generating concise codes and codes with more comments.

## 2 RELATED WORK

**Transformers for program synthesis.** Our work is based on Transformer for program synthesis (Roziere et al., 2020; Austin et al., 2021). Inspired by their capacities on a range of natural language tasks, modern transformer-based language models (Devlin et al., 2019; Radford et al., 2019; Raffel et al., 2020) have been adopted for program synthesis by treating programming languages in the same way as natural languages. A family of BERT-based Transformers are developed for code syntax (Kanade et al., 2020; Feng et al., 2020; Devlin et al., 2019; Guo et al., 2020). Later, CodeX (Chen et al., 2021a) and CodeT5 (Wang et al., 2021) adopted GPT2 (Radford et al., 2019) and T5 (Raffel et al., 2020), respectively, as backbones for both code understanding and generation. Different learning methods including learning from examples (Ellis et al., 2021) and neural-symbolic methods (Nye et al., 2020) were explored. Recently, AlphaCode (Li et al., 2022) combined large

transformer models pre-trained on massive program data with large-scale sampling, showing competitive performance in programming competitions. All these works mainly focused on training more powerful code-generation models and still used beam search (Graves, 2012) or sampling (Fan et al., 2018) during the Transformers' generation process.

**Test cases for program synthesis.** Our work is also related to using unit tests (Myers, 1979) for program synthesis. Tufano et al. (2020a;b) propose to generate test cases and corresponding accurate assert statements with Transformer models (Vaswani et al., 2017a). Recently, Roziere et al. (2022) leverage automatically generated unit tests to construct parallel training data for unsupervised code translation. Chen et al. (2018); Gupta et al. (2020); Chen et al. (2021b) directly synthesize domain-specific programs from input-output pairs without problem description. Ellis et al. (2019) use test cases to train a reinforcement learning agent and use a sequential Monte-Carlo sampling method for code generation. Unlike the prior work, we use unit-testing results as reward signals for a tree-search-based planning algorithm, which is further integrated with a Transformer-based large language model to generate better codes.

**Planning and reinforcement learning for code generation.** The code generation problem has been formulated as a sequential decision-making problem (Bunel et al., 2018; Chen et al., 2018), which enables designing and applying reinforcement learning (RL) and planning algorithms for code generation. RL has been used for both the training phase and the decoding phase of the Transformer for code generation. In the training phase, Le et al. (2022); Xu et al. (2019b) use an RL objective that optimizes the correctness of the generated programs instead of optimizing their similarity to the reference solutions. In the decoding phase, the Monte-Carlo tree search (MCTS) algorithm has been applied to search for high-quality codes. However, MCTS itself is unable to scale to larger domains. It is only used to generate domain-specific languages or for restricted programming synthesis tasks like assembly code generation (Xu et al., 2019a), Java bytecode translation (Lim & Yoo, 2016), and robot planning (Matulewicz, 2022). These tasks have a much smaller scale than generating Python codes in our work. Therefore, their frameworks do not require a large language model and do not need to address the challenge of integrating a planning algorithm with a large language model.

MCTS is more efficient and applicable to large domains when combined with deep learning or with a default policy as prior knowledge (Gelly & Silver, 2011; Silver et al., 2016; Simmons-Edler et al., 2018). We consider the same overall recipe in our work. Specifically, we contributed to integrating the large language model with a tree search algorithm to design a novel algorithm that is capable of solving competitive programming problems, and designed mechanisms to improve its efficiency.

**Planning in natural language generation.** Planning algorithms like MCTS have also been used to find the optimal text outputs for different natural language processing (NLP) tasks. For example, Scialom et al. (2021); Leblond et al. (2021); Chaffin et al. (2022) use pre-trained discriminators or pre-defined metrics as reward functions. We want to emphasize that we are the first to combine a tree search algorithm with large language models for general-purpose programming language generation. We design the interfaces between these two components and deal with the unique challenges of making the framework computationally efficient. Concurrently, other search algorithms like A* algorithm (Hart et al., 1968) are also applied in the Transformer decoding process. Lu et al. (2021) consider the constrained text generation problem and integrate A* with the beam search algorithm. However, their constraints are expressed as a formal logic expression, which is different from maximizing the pass rate in our problem.

## 3 METHOD

### 3.1 OVERVIEW

We consider the code generation problem for competitive programming, illustrated in Fig. 1. An agent is given the natural language description of a coding problem. It requires the agent to understand the problem description and generate a program that solves the problem. Similar to Li et al. (2022); Chen et al. (2018); Ellis et al. (2019), we assume that the agent has access to a set of test cases, where a test case is a pair of input, output strings. Given the input string, the agent is expected to generate a program that produces an output that exactly matches the test case's output string. The objective is to generate a program that passes the most number of test cases. To determine if the agent is able to generate programs that generalize to unseen test cases, we divide the test cases into *public* test cases and *private* test cases, following the terms in Li et al. (2022). The agent can only

---

**Problem Statement**

*Given is a string $S$. Replace every character in $S$ with $x$ and print the result.*

**Constraints**

*(1). $S$ is a string consisting of lowercase English letters.*

*(2). The length of $S$ is between $1$ and $100$ (inclusive).*

**Input**

*Input is given from Standard Input in the following format: $S$*

**Output**

*Replace every character in $S$ with $x$ and print the result.*

**Sample Test Input**

*sardine*

**Sample Test Output**

*xxxxxxx*

```
1 s=input()
2 s=list(s)
3 for i in range(len(s)):
4     for j in range(len(s)):
5         if s[i]=="x":
6             s[i]=j
7     print("".join(s))
8
```
Beam Search (Pass Rate: 0.00).

```
1 s=input()
2 s=list(s)
3 for i in range(len(s)):
4     if s[i]=="x":
5         s[i]="x"
6     else:
7         continue
8 print("".join(s))
```
Sampling + Filtering (Pass Rate: 0.22).

```
1 s=str(input())
2 for i in range(len(s)):
3     if s[i]!="x":
4         s=s[:i]+"x"+s[i+1:]
5
6 print(s)
7
8
```
PG-TD (Pass Rate: 1.00).

---

Figure 1: A code generation example for competitive programming, with the problem description (top) and the programs generated by baseline algorithms and our PG-TD algorithm (bottom).

access the public test cases during the program generation process, while we use the private test cases to evaluate the programs it generates.

Transformer models (Li et al., 2022; Hendrycks et al., 2021) have been widely applied for code generation thanks to their capacity in sequence-to-sequence modeling. In the Transformer's generation process, beam search (Graves, 2012) and sampling (Fan et al., 2018; Dabre & Fujita, 2020) are adopted to generate code sequences. However, these algorithms cannot easily optimize an objective different from what it is trained on (usually the similarity to the reference solutions). So we cannot directly use these generation algorithms to generate programs aiming to pass more test cases.

On the other hand, a planning algorithm can directly optimize the pass rate or any desirable programming objective. To use a planning algorithm, we follow Bunel et al. (2018); Ellis et al. (2019) to formulate the code generation problem as a Markov decision process (MDP) (Sutton & Barto, 2018). In this formulation, a **state** $s$ is the concatenation of the problem description and a partial or complete program, where a complete program ends with a special terminal token. An **action** $a$ is a token in the vocabulary set of the Transformer. There is a special termination action (the terminal token) that indicates that the agent believes the program is complete. The **transition function** deterministically concatenates a state $s$ with a token $a$, and an episode ends when the agent takes the termination action. The **reward** of state $s$ is the pass rate of the program on the public test cases when $s$ is a complete program (*i.e.,* when the last token of $s$ is the terminal token). The reward of a partial program is always 0.

There is rich literature on finding the optimal policy in an MDP. In this paper, we consider a tree search-based planning algorithm inspired by Monte-Carlo tree search (MCTS), illustrated in Fig. 2. Intuitively, the tree search algorithm maintains a tree structure where nodes correspond to states and edges correspond to actions. The algorithm starts from the root node (the initial state) and searches the state space to find terminal states with high rewards. It maintains 1) the number of times each node is visited and 2) a value function that maintains the maximum reward obtained by starting in node (or state) $s$ and taking action $a$. The algorithm would visit and expand nodes with either higher values (as they lead to higher-quality programs) or with smaller visit numbers (as they are under-explored). In the following part of this section, we describe how we integrate the tree search algorithm in the generation process of a Transformer.

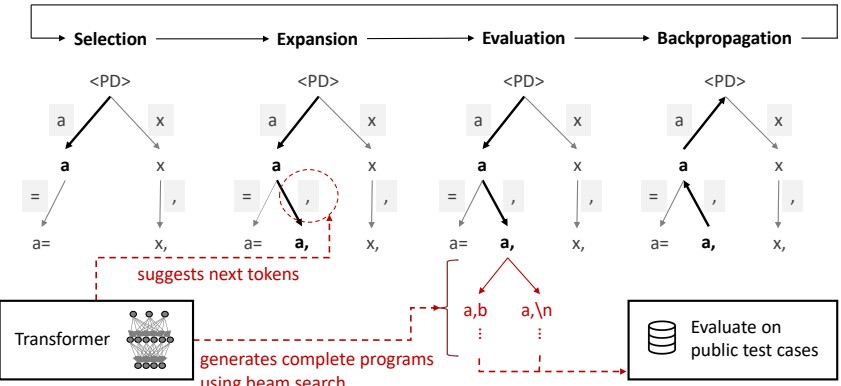

Figure 2: Illustration of using the Monte Carlo tree search algorithm in the Transformer generation process for code generation. <PD> stands for problem description.

## 3.2 PLANNING-GUIDED TRANSFORMER DECODING

Now we are ready to answer the question we asked in the introduction: Can we integrate a planning algorithm with a pre-trained code generation Transformer to generate better programs? We design a Transformer generation algorithm where a tree search algorithm is used to perform lookahead planning. The tree search algorithm alone may not be able to find high-quality codes due to the large search space. So the conventional Transformer beam search algorithm and the next-token probabilities provided by the pre-trained Transformer are used by the tree search algorithm to guide the search process.

We provide the pseudocode of our Planning-Guided Transformer Decoding algorithm (PG-TD) in Algorithm 1 and illustrate the whole process in Figure 2. The PG-TD algorithm follows the same framework as the standard MCTS algorithm, based on the implementation used in Silver et al. (2017). Here, we focus on how the Transformer is used in the tree search steps. We provide more details of our algorithm in Sec. D.1.

---

**Algorithm 1** The PG-TD algorithm.

---

**Require:** $root$: the current state; $c$: P-UCB exploration parameter; $k$: the maximum number of children of any node; $b$: the number of beams for Transformer beam search.
1: $program\_dict = \text{DICTIONARY}()$
2: **for** $i \leftarrow 1, 2, \ldots, max\_rollouts$ **do**
3:      $node \leftarrow root$
4:      # Selection
5:      **while** $|node.children| > 0$ **do**
6:          $node \leftarrow \text{P\_UCB\_SELECT}(node.children, c)$
7:      **end while**
8:      # Expansion
9:      $next\_tokens \leftarrow \text{TOP\_K}(node, k)$
10:     **for** $next\_token \in next\_tokens$ **do**
11:         $next\_state \leftarrow \text{CONCAT}(node, next\_token)$
12:         Create node $new\_node$ where $new\_node \leftarrow next\_state$
13:         Add $new\_node$ to the children of $node$
14:     **end for**
15:     # Evaluation
16:     $p \leftarrow \text{BEAM\_SEARCH}(node, b)$
17:     $r \leftarrow \text{GET\_REWARD}(p)$
18:     $program\_dict[p] = r$
19:     # Backpropagation
20:     Update and the values of $node$ and its ancestors in the tree with $r$
21: **end for**
22: **return** program in $program\_dict$ with the highest reward

---

In the **selection** step, we follow Silver et al. (2017) and use the P-UCB algorithm to select which branch of the tree we want to explore. In P-UCB, we weigh the exploration term by the probability of the next tokens determined by the Transformer. So the tree search selects higher-probability tokens more often. The selection algorithm is parameterized by an exploration parameter, $c$, where a higher $c$ value leads to more exploration. We describe the details of P-UCB in Sec. D.1.

In the **expansion** step, after a node in the tree is selected, we select the possible next tokens and add the corresponding next states as new nodes to its children list (for succinctness, a node also refers to the state that it represents). Sampling a random token as in the standard MCTS may very likely cause a syntax error. So we call TOP_K to get the most likely next tokens, where $\text{TOP\_K}(s, k)$ returns the $k$ most likely next tokens starting from $s$; $k$ is the maximum number of children that any node may have. The corresponding $k$ next states are the concatenations of the current state with each of

the next tokens suggested by the Transformer. These next states are added to the children list of the current node. (Line 9-14)

In the **evaluation** step, we need to evaluate the selected *node*. Note that *node* may still be a partial program. We cannot directly evaluate the quality of a partial program as we do not know how it will be completed and how many test cases it will pass. Here, we use the Transformer again by calling the BEAM_SEARCH function to generate a complete program from the current node, where BEAM_SEARCH($s, b$) generates a sequence using the Transformer beam search algorithm with the prefix $s$ and beam size $b$. We run the generated program on the public test cases to get its reward, and set it to be the value of *node* (Line 16-17). This value is backpropagated up in the tree so that the values of its ancestors are updated (Line 20).

**Information sharing between Transformer and tree search.** A keen reader may notice that if we follow the algorithm described above, there may be a lot of repeated computations. The key observation is that the Transformer beam search algorithm also *implicitly builds a tree structure*, which can be used by future iterations of tree search. In the rest of this section, we describe how we improve the algorithm's efficiency by sharing information in the Transformer beam search algorithm with tree search.

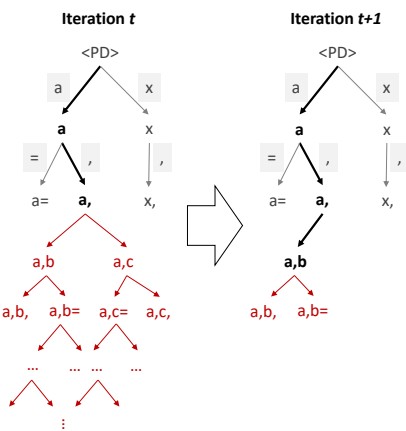

Figure 3: Illustration for caching in the PG-TD algorithm. The tree search part is visualized in black color and the Transformer beam search part is in red color.

Consider the example in Fig. 3, which shows two iterations of PG-TD. In the evaluation step of the $t$-th iteration, the Transformer beam search algorithm implicitly builds a tree to find the most likely sequences within a beam (Fig. 3 (left)). Because we only keep $b$ partial programs in the beam, it is a tree where only $b$ nodes with the highest likelihood are expanded at each level (in the illustration, $b = 2$). Other nodes are dropped and no longer considered by the beam search algorithm. In the $(t+1)$-st iteration, if the tree search algorithm selects "a,", such a state is already visited in the Transformer beam search in the $t$-th iteration. When the algorithm needs to find the top-k most likely next tokens, such information is already obtained in the $t$-th iteration and can be reused without recomputation. In our implementation, we cache the tree structure generated by the Transformer beam search algorithm. We call this implementation **tree structure caching**.

We can also cache the complete programs generated during the PG-TD evaluation step. In the evaluation step of the $t$-th iteration (Fig. 3), suppose the greedily-optimal sequence is "a,b=...". In the $(t + 1)$-st iteration, to generate a sequence starting with "a,b", the Transformer beam search algorithm will generate the same sequence "a,b=..." as before. (This is not necessarily true when the beam size is larger than 1. We will clarify this in Sec. D.2.) To improve the efficiency of BEAM_SEARCH, we cache the sequences that have been generated during the evaluation step. In the evaluation step of future iterations, PG-TD will check if the current state matches the prefix of any sequence that has been generated before, and use the generated sequence directly without calling the Transformer beam search function. We call this implementation **sequence caching**. We will empirically confirm the effectiveness of using these caching methods in the next section.

## 4  EMPIRICAL EVALUATION

In this section, we empirically examine the effectiveness and efficiency of our PG-TD algorithm by answering the following questions. **Q1:** Does PG-TD generate better programs than using the Transformer beam search algorithm and other competing baseline methods? Is our algorithm model-agnostic, showing improvement when applied to different Transformer models? **Q2:** Is the tree search algorithm an effective planning algorithm in PG-TD? Is it better than other planning or sampling-based algorithms? **Q3:** Is PG-TD efficient in terms of the number of times it runs the Transformer beam search algorithm and also the computation time it consumes? **Q4:** Are the caching methods effective in saving computation time? **Q5:** Can we use the samples generated by PG-TD in its search process and finetune the Transformer to generate even better solutions?

|  |  | Pass Rate (%) | | | | Strict Accuracy (%) | | | |
|---|---|---|---|---|---|---|---|---|---|
|  |  | APPS Intro. | APPS Inter. | APPS comp. | CodeContests | APPS Intro. | APPS Inter. | APPS comp. | CodeContests |
| APPS GPT-2 | Beam Search | 11.95 | 9.55 | 5.04 | 5.10 | 5.50 | 2.10 | 1.00 | 0.00 |
|  | Sampling+Filtering | 25.19 | 24.13 | 11.92 | 20.40 | **13.80** | 5.70 | 2.30 | 3.64 |
|  | SMCG-TD | 24.10 | 21.98 | 10.37 | 17.47 | 11.70 | 5.50 | 2.10 | 4.24 |
|  | PG-TD ($c = 4$) | **26.70** | **24.92** | **12.89** | **24.05** | 13.10 | **6.10** | **3.10** | **4.85** |
| APPS GPT-Neo | Beam Search | 14.32 | 9.80 | 6.39 | 5.73 | 6.70 | 2.00 | 2.10 | 0.00 |
|  | Sampling+Filtering | 27.71 | 24.85 | 12.55 | 25.26 | **15.50** | 5.80 | 3.00 | 4.24 |
|  | SMCG-TD | 25.09 | 20.34 | 9.16 | 15.44 | 13.80 | 5.10 | 1.80 | 3.03 |
|  | PG-TD ($c = 4$) | **29.27** | **25.69** | **13.55** | **26.07** | **15.50** | **6.43** | **3.50** | **4.85** |

Table 1: Results of PG-TD and other algorithms. The maximum number of Transformer generations for all algorithms is 256.

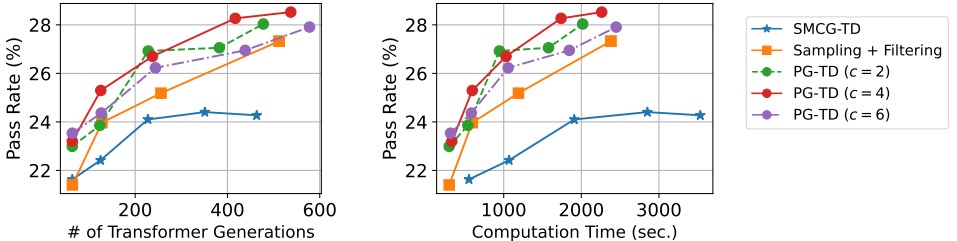

Figure 4: Pass rates of PG-TD and baseline algorithms vs. the number of Transformer generations (left) and the computation time (middle) on the introductory problems in the APPS test dataset (1000 problems), using the APPS GPT-2 Transformer model.

**Datasets and models.** Recently, several competitive programming datasets have been made available for benchmarking code generation algorithms. For each programming problem, they include the natural language program descriptions, sample solutions, and test cases. We evaluated PG-TD and the baseline algorithms on some popular benchmark datasets: APPS (Hendrycks et al., 2021) and CodeContests in AlphaCode (Li et al., 2022). The APPS dataset does not specify public vs. private test cases. So we split all the test cases of a program evenly into two sets, where the first set is used as the public test cases for the algorithms to optimize the pass rate, and the second set is used as the private test cases for evaluating the generated programs. For CodeContests, we use their public and generated test cases as our public test cases, and their private test cases as our private test cases. To show that PG-TD is model-agnostic and can be applied to different pre-trained Transformers and achieve a better performance, we use two popular pre-trained code-generation Transformers in the literature: GPT-2 and GPT-Neo finetuned on the APPS training dataset (Hendrycks et al., 2021).

**Algorithms.** We compare PG-TD with the following algorithms. **Beam search** only uses the Transformer beam search algorithm to generate the whole program, without using the test cases. This is the method used in Hendrycks et al. (2021); Chen et al. (2021a). We use the beam size of 5 in the Transformer generation function, which is the same choice as in Hendrycks et al. (2021).

We also implemented two baselines that use Transformer to sample programs and filter them using the test cases. **Sampling + Filtering (S+F)** generates a set of programs using the Transformer sampling algorithm. Once the programs are all generated, it computes the pass rates of all the programs on the public test cases and returns the program with the highest pass rate. To avoid generating low-probability tokens that may fail the program completely, we use top-3 sampling, that is, the Transformer only samples the token that is in the top-3 most-likely tokens in each step. The sampling temperature is 1. This method is similar to the algorithm in AlphaCode (Li et al., 2022), except that we did not perform clustering as we generate a much smaller number of programs.

To determine the effectiveness of the tree search algorithm, we considered another baseline, **Sequential Monte-Carlo-Guided Transformer Decoding (SMCG-TD)**, that replaces the tree search algorithm component with a sequential Monte-Carlo algorithm (Ellis et al., 2019). It is an iterative algorithm that maintains a population of partial programs. It determines the *fitness* of a partial program using the Transformer in the same way as the evaluation step in PG-TD. Partial programs with higher fitness scores are more likely to be selected in the next iteration. We leave more details about the baseline algorithms in Sec D.3.

For PG-TD, we set the maximum number of children of any node ($k$) to be 3, and the beam size ($b$) to be 1 by default. For the baseline methods, we sample at most 512 programs in Sampling

| Method | Time (sec.) |
|---|---|
| With both caching methods | **1312.27** |
| W/ only sequence caching | 1430.63 |
| W/ only tree structure caching | 1899.17 |
| W/o caching | 2206.38 |

Table 2: Affects of using caching for PG-TD on the first 100 problems on the APPS introductory dataset, using the APPS GPT-2 (1.5B).

| Method | Pass Rate (%) | Strict Acc. (%) |
|---|---|---|
| Original | 14.32 | 6.70 |
| FT w/ S+F | 15.24 | **7.90** |
| FT w/ PG-TD | **16.54** | **7.90** |

Table 3: Performance on APPS introductory dataset with finetuned (FT) APPS GPT-2 (2.7B), using Beam Search for code generation.

+ Filtering, and maintain a set of 200 partial programs in each iteration for SMCG-TD. We will use these parameters for the following experiments unless noted otherwise. The experiments are conducted on the test set of the APPS dataset (5000 problems) (Hendrycks et al., 2021) and the test set of CodeContests (165 problems) (Li et al., 2022). We use the same metrics as in Hendrycks et al. (2021), which are *pass rates* and *strict accuracies* on the private test cases. Specifically, the pass rate is the average percentage of the private test cases that the generated programs pass over all the problems. The strict accuracy is the percentage of the problems where the generated programs pass all the private test cases.

**Effectiveness of PG-TD.** To make a fair comparison, we evaluate the best programs found by the algorithms when they use the same *number of Transformer generations*. Specifically, the number of Transformer generations is the number of function calls of BEAM_SEARCH in PG-TD and SMCG-TD (when no cached sequences are found and used in sequence caching), and the number of sampling function calls in S+F.

As shown in Table 1, PG-TD consistently outperforms all the other baselines on all the datasets under the pass rate metric. As we optimize the pass rate, we expect our algorithm to outperform other baselines under strict accuracy as well. This is almost true except that our algorithm is matched or outperformed by S+F on the APPS introductory set, mostly due to the fact that this set of programs is less challenging. The gaps between different algorithms are also small on this set. Overall, these results confirm that our algorithm indeed generates programs that pass more test cases than the baseline methods (answering **Q1**). Specifically, S+F and SMCG-TD both use test cases to filter programs generated by the Transformer, while their performance is overall outperformed by PG-TD. So the tree search-based planning algorithm is indeed powerful enough and can be effectively integrated with Transformer (answering **Q2**). We also report the performance of PG-TD and S+F under the **n@k** and **pass@k** metrics (Li et al., 2022) in Sec. A (Table 6).

**Efficiency of PG-TD.** To show PG-TD's efficiency, we report the pass rates of the best programs found by the algorithms using the same computation time (Fig. 4 (right)). The experiments are run on the introductory problems in the APPS test set. Since one may implement these algorithms differently (possibly using parallelization), we also report the results using the number of Transformer generations as a budget (Fig. 4 (left)).

For PG-TD, we set $k = 3, b = 1$ and vary the P-UCB exploration parameter $c$ to be 2, 4, and 6. We see that PG-TD with $c = 4$ has the best performance, while setting $c = 2$ tends to under-explore and setting $c = 6$ over-explores. With the same computational budget, S+F generates programs with lower pass rates (answering **Q3**). This confirms our observation that S+F, like the algorithm used in AlphaCode (Li et al., 2022), generates solutions only according to the Transformer's generation probabilities, without taking their pass rates into consideration until all the programs are generated. PG-TD actively considers the pass rates of the generated programs during the generation process, which achieves better efficiency. On the other hand, SMCG-TD also uses the Transformer and public test cases to guide the generation process. However, due to its sampling nature, it cannot do multi-step lookahead search as in PG-TD, which results in worse performance. We leave additional results of varying $k, b$ for PG-TD and varying temperatures for S+F in Sec. A.

**Effectiveness of caching.** In terms of the design choices of tree structure caching and sequence caching, we performed ablative studies to verify their efficiencies. In Table 2, we compare versions of PG-TD with and without tree structure caching and sequence caching. As we expect, without sequence caching, the algorithm needs to regenerate whole sequences, ending up consuming much more time. Without tree structure caching, the algorithm is slightly slower as it needs to call Transformer to get the most likely next tokens (answering **Q4**).

**Finetuning transformer with PG-TD-generated samples.** Since we are able to generate solutions with high pass rates using our PG-TD algorithm, can we use these generated solutions to finetune the code generation Transformers to further improve their performance? This may effectively solve the problem that high-quality human-written programs that can be used to train the code generation Transformers are scarcely available. Concretely, we first run PG-TD on the training set of APPS. We then add the generated solutions with pass rates larger than 80% to the APPS sample solution set, and use the augmented solution set to finetune the GPT-2 (2.7B) model. With the finetuned Transformer model, we run beam search to generate solutions on the test set to see if it has a better performance. We use the first 500 problems in the interview-level problems in APPS test set for validation and the introductory-level problems in APPS test set for testing. The results are reported in Table 3. After finetuning with PG-TD-generated solutions, we see improvement in both pass rate and strict accuracy. More details are in Sec. D.4.

**Optimizing other code generation objectives.** Beyond the pass rate, we can make the algorithm versatile by using different reward functions. We consider two new objectives, *code length penalty* and *comment encouragement*. As shown in Table 4, the code length penalty reward function makes the model generate more concise codes; the comment encouragement reward function encourages models to generate codes with more comments. Both objectives still achieve reasonable pass rates on the public test cases. We provide qualitative examples and more details of the reward functions in Sec. C of the appendix.

| Methods | Code length$\downarrow$ | Comment number $\uparrow$ | Pass rate $\uparrow$ |
|---|---|---|---|
| Default | 248.42 | 0.68 | **23.14** |
| Length Penalty | **190.73** | - | 22.82 |
| Comment Encouragement | - | **3.11** | 21.65 |

Table 4: Performance of controllable code generation. Code length denotes the length of the generated code string and comment number denotes the number of code lines that contains comments.

**Using automatically-generated test cases.** The datasets we use in this paper both contain test cases that can be used to compute rewards for PG-TD. However, in reality, human-specified test cases are not always available. Recently, Chen et al. (2022) observe that Transformer pre-trained on code generation can also generate useful test cases by adding an `assert` keyword at the end of the prompt. We follow the prompt design in Chen et al. (2022) to automatically generate test cases and run our PG-TD algorithm using the automatically-generated test cases. Empirically, we confirm that compared with beam search, PG-TD still has a higher strict accuracy by using automatically-generated test cases to verify the generated programs. We provide the details of the method and the experiments are in Sec. B.

## 5 DISCUSSION AND CONCLUSION

In summary, we proposed a novel algorithm that uses the power of a pre-trained Transformer and a tree search algorithm inspired by Monte-Carlo tree search. We evaluate our algorithm and empirically show that our algorithm generates programs of higher quality compared with competing baseline algorithms in different settings. We also design model structure-specific caching mechanisms which contribute to saving computational expenses. We show that our algorithm is versatile and can generate codes under objectives other than the pass rate without finetuning the Transformer. We hope our work can inspire more ways of incorporating planning algorithms into the Transformer generation process for code generation or other problems.

One limitation of our algorithm is its reliance on test cases, although we find that even a small number of test cases can help find better solutions (Sec. A, Table 7). To address this limitation, we provided results to show that our algorithm can take advantage of automatically-generated test cases (Sec. B). Our algorithm is also more computationally expensive than the pure Transformer beam search algorithm since ours needs to run the beam search algorithm multiple times within the tree search algorithm. In the future, we consider improving the framework's performance by using a value function to estimate the programs' pass rates, similar to the evaluation function in Silver et al. (2016) for mastering the game of Go. We can learn a neural state representation for partial programs as in Chen et al. (2018). We also consider using parallel tree search algorithms (Chaslot et al., 2008) for the evaluation step to parallelize the computation for different states.

**Acknowledgements.** This work was supported by the MIT-IBM Watson AI Lab, DARPA MCS, and gift funding from MERL, Cisco, and Amazon. We would also like to thank the computation support from AiMOS, a server cluster for the IBM Research AI Hardware Center.

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

APPENDIX

In this appendix, we supplement the main paper by providing more thorough empirical evaluations to back up our claims and more detailed descriptions of the algorithms to help readers better understand our paper.

This appendix is organized as follows.

- In Sec. A, we provide more comprehensive results of our algorithm and the baseline algorithms. We also include the license information of the datasets we use.

- In Sec. B, we consider the scenario where test cases are not provided. We evaluate our PG-TD algorithm using automatically-generated test cases.

- In Sec. C, we provide empirical evidence for our claims in the discussion section that our algorithm is versatile and can be used to optimize different code generation objectives other than the pass rate. We consider the objectives of generating concise codes and generating codes with more comments.

- In Sec. D, we provide more details on the components in PG-TD as well as the baseline algorithms.

- In Sec. E, we illustrate more examples of the codes generated by our PG-TD algorithm and the baseline algorithms.

- In Sec. F, we discuss more on the advantages and the potential negative social impacts of our algorithm.

## A    EMPIRICAL EVALUATION

We reported the performance of our algorithm and other baselines on the whole APPS dataset (Hendrycks et al., 2021) and CodeContests (Li et al., 2022). The APPS test dataset contains coding problems at three levels: introductory (1000 problems), interview (3000 problems), and competition (1000 problems).

In addition to our results in Table 1 in the main paper, Table 5 shows the results where the budget of the number of Transformer generations is 512. As we expect, with a larger number of generated programs, the gap between PG-TD and the rest is smaller. Sampling + Filtering has a better strict accuracy than our algorithm on some subsets of datasets. We also experimented with other parameter settings of PG-TD. In Fig. 5 and 6, we vary the beam size ($b = 1, 3, 5$) under $c = 2, 4$. In Fig. 7, we vary the maximum number of children of any node ($k = 2, 3, 4$).

Although one may expect expanding the width of the tree (increasing $k$) can help find better programs, considering generating less-likely tokens suggested by the Transformer may not contribute to finding better programs. In fact, it wastes the budget of the number of Transformer generations and the computation time by considering less likely tokens. On the other hand, increasing the beam size ($b$) does help improve the pass rate when the number of rollouts (corresponding to the number of Transformer generations) is small. However, increasing the beam size under a larger number of rollouts does not always increase the pass rate (Fig. 7), which may be caused by an overfitting issue. Increasing $b$ also costs more computation time even though sequence caching is used.

In Fig. 8, we use different temperatures in the Sampling + Filtering algorithm ($t = 0.6, 0.8, 1$), confirming that our choice of $t = 1$ in the experiments in the main paper does have the best performance.

**n@k and pass@k metrics.**    We also report the performance of PG-TD and S+F under the **n@k** and **pass@k** metrics (Li et al., 2022). Given a problem, each algorithm is asked to generate **k** programs and submit **n** programs for evaluation (they submit the **n** programs with the highest reward on the public test cases). **n@k** is the proportion of the problems where any of the **n** submitted programs passes all the private test cases. **pass@k** is the proportion of the problems where any of the **k** generated programs passes all the private test cases (effectively **k@k**). For PG-TD, the **k** samples are the complete programs found in the first **k** rollouts. The results are reported in Table 6, evaluated on the APPS introductory problems. We observe that for smaller **n** or **k** values, PG-TD finds programs

| | | Pass Rate (%) | | | | Strict Accuracy (%) | | | |
|---|---|---|---|---|---|---|---|---|---|
| | | APPS Intro. | APPS Inter. | APPS comp. | CodeContests | APPS Intro. | APPS Inter. | APPS comp. | CodeContests |
| APPS GPT-2 | Beam Search | 11.95 | 9.55 | 5.04 | 5.10 | 5.50 | 2.10 | 1.00 | 0.00 |
| | Sampling+Filtering | 27.33 | 25.75 | 12.09 | 21.57 | **14.80** | **6.97** | 2.40 | 4.24 |
| | SMCG-TD | 24.40 | 22.18 | 10.60 | 17.72 | 11.80 | 5.87 | 2.50 | 4.24 |
| | PG-TD ($c = 4$) | **28.27** | **26.13** | **13.74** | 25.91 | 14.40 | 6.63 | **4.00** | **4.85** |
| APPS GPT-Neo | Beam Search | 14.32 | 9.80 | 6.39 | 5.73 | 6.70 | 2.00 | 2.10 | 0.00 |
| | Sampling+Filtering | 30.23 | 26.58 | 13.12 | 26.57 | **16.70** | **7.40** | 3.10 | 4.24 |
| | SMCG-TD | 24.77 | 20.52 | 9.19 | 15.42 | 14.00 | 5.20 | 1.70 | 2.42 |
| | PG-TD ($c = 4$) | **30.38** | **26.89** | **13.28** | 27.10 | **16.70** | 6.90 | **3.20** | **5.45** |

Table 5: Results of PG-TD and other algorithms. The maximum number of Transformer generations for all algorithms is 512.

| | 1@10 | 1@50 | 1@100 |
|---|---|---|---|
| PG-TD ($c = 4$) | **8.1** | **10.2** | **11.6** |
| S+F | 6.5 | 9.7 | 11.4 |
| | 5@10 | 5@50 | 5@100 |
| PG-TD ($c = 4$) | **12.5** | **14.9** | 16.2 |
| S+F | 11.9 | 14.6 | **18.2** |
| | pass@10 | pass@50 | pass@100 |
| PG-TD ($c = 4$) | **14.7** | **23.7** | 27.3 |
| S+F | **14.7** | 23.2 | **28.3** |

Table 6: **n@k** and **pass@k** results of PG-TD and Sampling, evaluated on the APPS introductory problems.

| | Pass Rate | Strict Accuracy |
|---|---|---|
| 0 public test cases (the beam search baseline) | 13.61 | 4.2 |
| 1 public test case | 23.27 | 7.4 |
| 3 public test cases | 33.02 | 13.4 |
| 5 public test cases | 32.99 | 14.2 |
| Half of all test cases (10.44 public test cases on average) | 37.71 | 14.8 |

Table 7: The performance of PG-TD using different numbers of public test cases, evaluated on the first 500 APPS introductory problems.

that are more likely to pass all the private test cases. When more programs are generated (**k**) or more programs are submitted for evaluation (**n**), the performance of PG-TD is matched or outperformed by S+F.

**Using different numbers of public test cases.** On the APPS dataset, we split all the test cases evenly into public test cases and private test cases. To investigate how many test cases are enough for PG-TD to achieve a satisfactory performance, we consider using 1, 3, and 5 public test cases and using the rest of the test cases as private test cases. The results are reported in Table 7, evaluated on the first 500 APPS introductory problems. We observe that even with a small number of public test cases, PG-TD is able to find programs with higher a pass rate and strict accuracy than beam search. This also suggests that the Transformer can serve as a regularizer in code generation. With a small number of public test cases, PG-TD still generates programs similar to human-written programs without overfitting the pubic test cases.

**Results on Codex.** To evaluate the effectiveness of our algorithms on more powerful code-generation Transformers, we run both beam search and PG-TD algorithms using Codex (Chen et al., 2021a) as the backbone. We follow the "best practices" provided in the Codex guide to put the problem description into a block comment in the beginning, and ask Codex to complete the rest. The prompt looks like the following.

```
"""
Python 3
{problem description}
"""
```

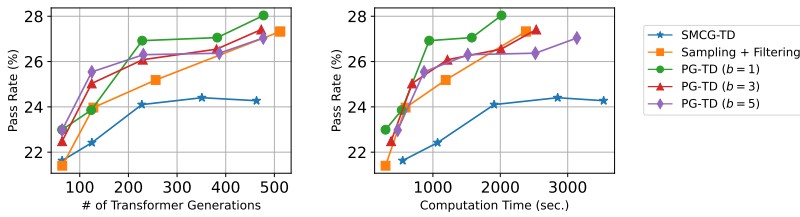

Figure 5: Results of PG-TD ($c = 2$) on the APPS introductory dataset, using the APPS GPT-2 Transformer model.

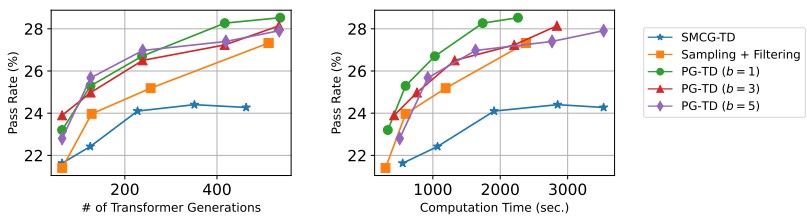

Figure 6: Results of PG-TD ($c = 4$) on the APPS introductory dataset, using the APPS GPT-2 Transformer model.

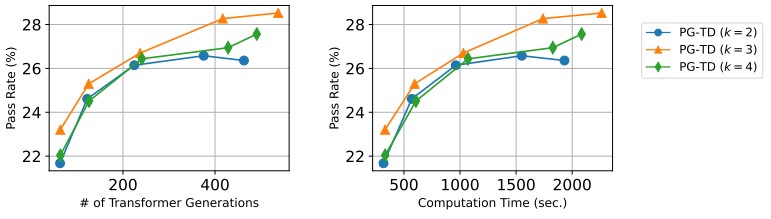

Figure 7: Results of PG-TD ($c = 4$) with different $k$ on the APPS introductory dataset, using the APPS GPT-2 Transformer model..

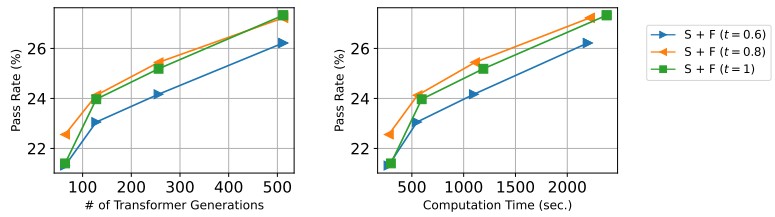

Figure 8: Results of Sampling + Filtering with different temperatures on the APPS introductory dataset, using the APPS GPT-2 Transformer model.

As we expect, PG-TD helps Codex generate better codes (Table 8). This further validates our claim that our method is model-agnostic and can help a pre-trained code generation Transformer generate better codes. Due to the limits of the OpenAI APIs, we are only able to test the algorithms on a subset of data. We also show a concrete example where PG-TD outperforms beam search in the following sections in Fig. 16.

**Using sampling-based tree search.** We could use a sampling approach in the evaluation step of PG-TD instead of using beam search, which would be a more faithful implementation of MCTS. We experimented with the two versions of PG-TD and find that overall using beam search indeed has a better performance (Table 9). The gap could be caused by that we only do sampling once to evaluate any node. We could estimate the value of nodes more accurately by sampling multiple times. However, it would make our algorithm more computationally expensive.

| Model | Decoding Algorithm | Pass Rate (%) | Strict Accuracy (%) |
|---|---|---|---|
| APPS GPT-2 (1.5B) | Beam Search | 11.65 | 2 |
| APPS GPT-2 (1.5B) | PG-TD | 36.42 | 6 |
| Codex (code-davinci-002) | Beam Search | 33.26 | 14 |
| Codex (code-davinci-002) | PG-TD | 59.14 | 31 |

Table 8: Comparison between using different Transformers (APPS and Codex) as the backbone for code generation, evaluated on the first 100 APPS introductory problems.

| Decoding Algorithm | Pass Rate (%) | | | Strict Accuracy (%) | | |
|---|---|---|---|---|---|---|
| # of Transformer Generations | 128 | 256 | 512 | 128 | 256 | 512 |
| PG-TD using Beam Search | 36.19 | 37.71 | 38.57 | 14.4 | 14.8 | 15.6 |
| PG-TD using Sampling | 33.95 | 35.67 | 37.87 | 13.8 | 15.4 | 17.2 |

Table 9: PG-TD using beam search vs. sampling in the evaluation step, evaluated on the first 500 APPS introductory problems.

| | Compilation Error (%) | Runtime Error (%) |
|---|---|---|
| APPS GPT-2 (1.5B), Beam Search | 5.58 | 32.95 |
| APPS GPT-2 (1.5B), Sampling + Filtering | 4.68 | 27.38 |
| APPS GPT-2 (1.5B), PG-TD | 1.93 | 19.5 |

Table 10: Failure case analysis. The percentages are averaged over the APPS introductory dataset.

| | Strict Accuracy (%) |
|---|---|
| Beam Search | 26.82 |
| PG-TD (using auto-generated test cases) | 74.53 |

Table 11: Comparison between beam search and TG-PD using automatically-generated test cases, using the Codex model and evaluated on the HumanEval dataset.

**Failure mode analysis.** In Table 10, we report the average percentages of test cases where compilation errors and runtime errors occur. We use the error analysis utility provided in the original APPS codebase. As we expect, PG-TD executes the generated programs and finds the ones with higher pass rates, so it dramatically reduces the percentage of both types of errors.

**Assets and licenses.** Our experiments are run on machines with two Intel(R) Xeon(R) Gold 6258R CPUs (@ 2.70GHz), and one V100-SXM2 GPU. The APPS dataset (Hendrycks et al., 2021) is released under MIT License [2]. The CodeContests dataset (Li et al., 2022) is released under Apache License 2.0 [3]. The Huggingface Transformer (Wolf et al., 2019) is released under Apache License 2.0 [4].

# B USING AUTOMATICALLY-GENERATED TEST CASES

When human-specified test cases are not available, our algorithm can still rely on automatically-generated test cases. We first follow Chen et al. (2022) to use the Codex model and evaluate the code generation algorithms on the HumanEval dataset (Chen et al., 2021a). We adopt the same procedure as Chen et al. (2022) to automatically generate test cases. Concretely, we use a prompt that includes the natural language description of the problem and an `assert` statement at the end of the prompt. We also removed the example input, output pairs in the problem description to avoid

---

[2] https://github.com/hendrycks/apps/blob/main/LICENSE
[3] https://github.com/deepmind/code_contests/blob/main/LICENSE
[4] https://github.com/huggingface/transformers/blob/main/LICENSE

the Transformer from directly copying the test cases in the problem description. The following is an example of the prompt for test case generation.

```
from typing import List

def has_close_elements(numbers: List[float], threshold: float) -> bool:
    """ Check if in given list of numbers, are any two numbers closer to
    each other than given threshold.
    """
    pass

# check the correctness of has_close_elements
def check(candidate):
    assert candidate
```

**Does Codex generate correct test cases?** In the HumanEval dataset, the generated solutions are evaluated by the test script by calling the check function. So we use the strict accuracy (pass@1) metric that counts the percentage of the problems where the generated solutions pass the check function. To evaluate the quality of the automatically-generated test cases, we compute the strict accuracies of the sample solutions (correct solutions written by human programmers) on these test cases. Clearly, the strict accuracies of the sample solutions on the *ground-truth* test cases should be 100%. We evaluate the sample solutions on the automatically-generated test cases and find the corresponding strict accuracy is 72.56%, which confirms that the automatically-generated test cases are mostly correct.

**Can PG-TD take advantage of the automatically-generated test cases?** We report the average strict accuracy of the generated programs on a subset of HumanEval problems in Table 11. We confirm that the strict accuracy of PG-TD is higher as it uses high-quality automatically-generated test cases to verify the generated programs.

## C   PLANNING FOR OTHER CODE GENERATION OBJECTIVES

Besides the default reward function that optimizes the pass rate, we show the versatility of the proposed algorithm by setting two new objectives, code length penalty and comment encouragement.

**Code length penalty**. We make the planner generate more concise codes by using the following reward function

$$\mathcal{R}_{\text{length}} = p + \lambda_1 \times e^{-l_c/t}, \tag{1}$$

where $p$ is the average pass rate on the public test case set and $l_c$ is the length of the code string. $\lambda$ and $t$ are hyperparameters that control the weight of the code length penalty and are set to $0.1$ and $20$, respectively. As shown in Figure 9, the generated solution becomes more concise and the code string length decreases from 187 to 78 while still passing all the test cases.

**Comment encouragement**. We can also generate solutions with more comments. We achieve this goal by using the following reward function

$$\mathcal{R}_{\text{comment}} = p + \lambda_1 \times e^{-l_c/t} + \lambda_2 \times \min(1, \frac{N_{cm}}{N_{max}}), \tag{2}$$

where $N_{cm}$ is the number of "#" in the code string and $\lambda_2$ is set to $0.2$, controlling the weights of the comment lines. If we simply add $\lambda_2 \times N_{cm}$ as the comment encouragement term, the planner would generate code with repeated meaningless comment lines to get more rewards. To handle this problem, we add the code length penalty term, $\lambda_1 \times e^{-l_c/t}$. We also upper-bound the rewards for the number of comment lines by setting the comment encouragement term to be $\lambda_2 \times \min(1, \frac{N_{cm}}{N_{max}})$, where $N_{max} = 5$ in the example shown in Figure 10. As shown in Figure 10, we can generate solutions with more comment lines with this designed reward function.

***Problem Statement***

*Takahashi loves palindromes. Non-palindromic strings are unacceptable to him. Each time he hugs a string, he can change one of its characters to any character of his choice. Given is a string S. Find the minimum number of hugs needed to make S palindromic.*

***Constraints***

 *(1). S is a string consisting of lowercase English letters.*

 *(2). The length of S is between 1 and 100 (inclusive).*

***Input***

 *Input is given from Standard Input in the following format:S.*

***Output***

 *Print the minimum number of hugs needed to make S palindromic.*

***Sample Test Input 1***

 *redcoder*

***Sample Test Output 1***

 *1*

*For example, we can change the fourth character to o and get a palindrome redooder.*

```python
# cook up a palindrome
s=input()
n=len(s)
c=0
if n%2==0:
        for i in range(n//2):
                if s[i]!=s[n-i-1]:
                        c+=1
        print(c)
else:
        for i in range(n//2):
                if s[i]!=s[n-i-1]:
                        c+=1
        print(c)
```

Default reward function (Pass Rate: 1.00, code string length: **187**).

```python
s=input()
ans=0
for i in range(len(s)//2):
        if s[i]!=s[~i]: ans+=1
print(ans)
```

Reward function with code length penalty (Pass Rate: 1.00, code string length: **78**).

Figure 9: A code generation example with code length penalty. The problem description is shown on the top. The programs generated by default PG-TD and PG-TD with code length penalty are shown at the bottom. We can generate more concise code by code length penalty reward.

---

***Problem Statement***
*A + B is often used as an example of the easiest problem possible to show some contest platform. However, some scientists have observed that sometimes this problem is not so easy to get accepted. Want to try?*
***Input***
    *The input contains two integers a and b $(0 \leq a, b \leq 10^3)$, separated by a single space.*
***Output***
    *Output the sum of the given integers.*
***Sample Test Input 1***
    5 14
***Sample Test Output 1***
    19
***Sample Test Input 2***
    381 492
***Sample Test Output 2***
    873

---

```
1  a, b = map(int, input().split())
2  print(a + b)
3
```

Default reward function.

```
1  #!/usrbin/env python3
2  # coding =utf-8 #py3 runs on py2
3  a,b=map(int,input().split())
4  print(a+b)
```

Reward function with comment encouragement (code with comments).

Figure 10: A code generation example with code comment encouragement. We can generate codes with more comments by using the code comment encouragement reward.

# D  ALGORITHM DETAILS

## D.1  PG-TD

In this section, we describe our PG-TD algorithm in more details. We show our framework in Fig. 12 again for convenience. The algorithm maintains a search tree structure where nodes correspond to states and edges correspond to actions. For each node $s$, it maintains the number of times that it has been visited, denoted by $s.visits$. For a state, action pair, $s, a$ (a stat, action pair corresponds to an edge in the tree), the algorithm maintains a function $Q(s, a)$, which is the maximum reward obtained by starting in $s$ and taking action $a$. Conventionally, $Q(s, a)$ is the *average* return. Since our code generation problem is deterministic (that is, there are no stochastic transitions), we find the performance is better if we keep track of the pass rate of the best program generated from a node.

The algorithm first **selects** a node for expansion. It starts from the root node and selects subtrees recursively until it reaches a node that has not been expanded before (with no children). A common node selection criterion is upper confidence bound (UCB) (Kocsis & Szepesvári, 2006), which leverages between exploiting known-to-be-good states and exploring less-visited states, We adapt the P-UCB heuristic used in Silver et al. (2017), which is a variation of UCB, as follows,

$$\text{P-UCB}(s, a) = Q(s, a) + \beta(s) \cdot P_{\text{Transformer}}(a|s) \cdot \frac{\sqrt{\log(s.visits)}}{1 + s'.visits}, \tag{3}$$

where $s'$ is the state (deterministically) reached by taking action $a$ in $s$; $P_{\text{Transformer}}(a|s)$ is the probability that token $a$ is the next token given the partial program $s$, which is provided by a Transformer model. $\beta$ is the weight for exploration, which depends on the number of visits of $s$, defined as

$$\beta(s) = \log(\frac{s.visits + c_{base} + 1}{c_{base}}) + c. \tag{4}$$

The P_UCB_SELECT function in Alg. 1 returns the action that has the highest P-UCB value,

$$\text{P\_UCB\_SELECT}(s, c) = \arg\max_a \text{P-UCB}(s, a). \tag{5}$$

Note that $c$ parameterizes the exploration weight in Eq. 4. We omit it in Eq. 5 for notational conciseness.

Intuitively, P_UCB_SELECT$(s, c)$ would return an action $a$ more often if 1) $Q(s, a)$ is large, which means that it has found high-quality programs starting from $s, a$; or 2) $P_{\text{Transformer}}(a|s)$ is large, which means that the Transformer believes that $a$ is a highly likely next token; or 3) $s.visits$ is large but $s'.visits$ is small, which means $s'$ is under-explored. We did a hyperparameter search and set $c_{base} = 10$, and used $c = 2, 4, 6$ in our experiments.

In the following steps, the algorithm **expands** the node by adding its children to the tree and **evaluates** the node using BEAM_SEARCH, as we described in the main paper. Finally, the reward of the completed program $r$ is **backpropagated** to its parents recursively until it reaches the root to update their $Q$ values. For all state, action pairs, $s'', a''$, along the trajectory in the tree from the root to the selected node, $Q(s'', a'')$ is updated accordingly using the new value: $Q(s'', a'') \leftarrow \max(Q(s'', a''), r)$.

**Comparison with the standard MCTS algorithm.**  It is worth noting that our implementation of MCTS in PG-TD is different from the standard MCTS algorithm. If we use the standard MCTS algorithm, it is computationally intractable to search the large space of possible programs to find high-quality ones.

For comparison, we visualize the standard MCTS algorithm in Fig. 11. In the expansion step, MCTS selects the next available action in the action set, and adds the state that can be reached by following the action. In the example, action "b" is taken, and the new state that is added to the tree is "ab". In the evaluation step, MCTS evaluates the new state by executing a random policy from the new state and computes the value of the policy.

It is impractical to apply the standard MCTS algorithm to domains with large state spaces or large action spaces, as we cannot afford to try all possible actions in the expansion step. A random policy in the evaluation step also has a high variance in estimating the value of the new state. In PG-TD, we use a pre-trained Transformer to resolve these limitations of the standard MCTS algorithm.

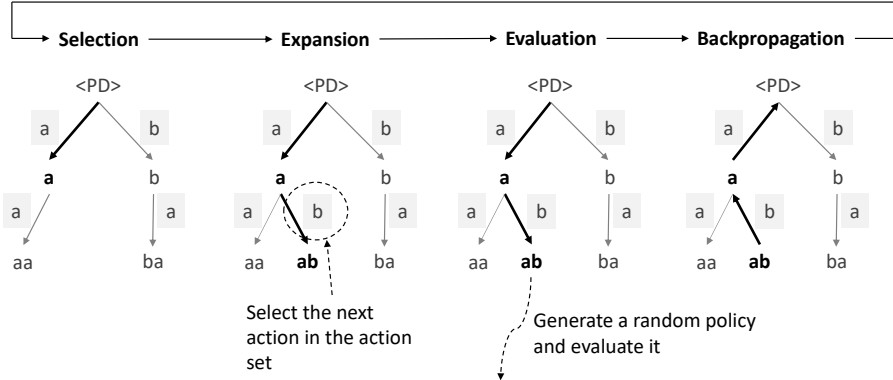

Figure 11: The standard Monte-Carlo tree search algorithm, without using a pre-trained Transformer. <PD> stands for problem description.

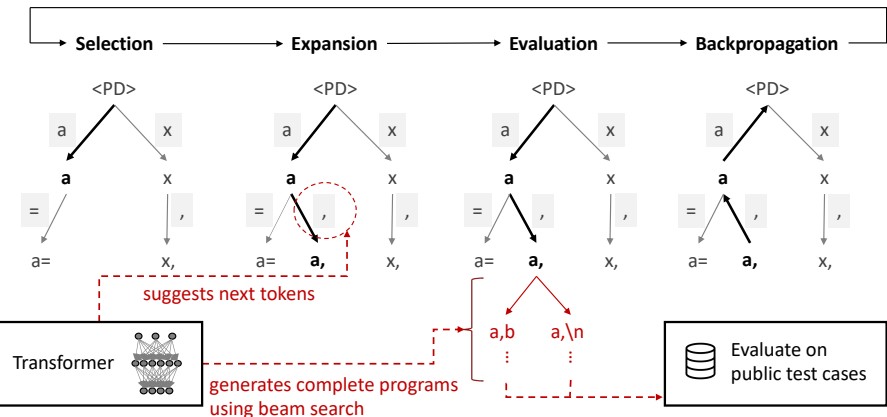

Figure 12: Our proposed framework, PG-TD, repeated here for convenience.

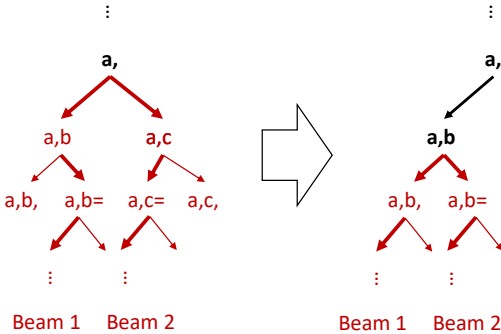

Figure 13: Illustration of an example where sequence caching may not return the correct sequence when $b > 1$.

## D.2 EFFICIENT IMPLEMENTATION BY INFORMATION SHARING

As discussed in the main paper, there may be calls to the Transformer generation functions the perform redundant computations. We have described tree structure caching in the main paper. We provide more details on sequence caching in this section.

---

**Algorithm 2** Sampling + Filtering.

---

**Require:** $sample\_num$: the number of samples to generate using Transformer.
  1: **for** $i \leftarrow 1, 2, \dots, sample\_nums$ **do**
  2:     $p_i \leftarrow \text{TRANSFORMER\_SAMPLE}(\langle PD \rangle, k = 5)$
  3:     $rewards[i] \leftarrow \text{GET\_REWARD}(p_i)$
  4: **end for**
  5: **return** $p_i$ with the largest $rewards[i]$

---

**Sequence caching.** The goal of sequence caching is to reduce the computational cost of the evaluation step of PG-TD. In the evaluation step, the Transformer beam search algorithm is called to generate a complete program from the current node. These complete programs and their rewards are both cached and used in a future iteration.

However, when $b > 1$, using cached sequences in the previous iterations may be suboptimal. Let's consider the example in Figure 13, where $b = 2$. Starting from "a," (the left figure), suppose the beam search algorithm returns a sequence by searching two beams with the prefixes of "a,b" and "a,c", respectively (the bold lines). In the next iteration (the right figure), to evaluate "a,b" using the beam search algorithm, we may not use the sequence found in the previous iteration, which *only searches the subtree of "a,b" with one beam*. In other words, we may find a sequence with a higher likelihood by re-running the beam search algorithm with $b = 2$ starting from "a,b".

Although using sequence caching with $b > 1$ may underestimate the values of programs in the evaluation step in PG-TD, we still use sequence caching for all choices of $b$ for the merits of computational saving.

### D.3    BASELINE ALGORITHMS

**Sampling + Filtering.** We have described the sampling algorithm in the main paper. The pseudocode is shown in Alg. 2.

**Sequential Monte-Carlo-Guided Transformer Decoding.** Sequential Monte-Carlo-Guided Transformer Decoding (SMCG-TD) is another algorithm that we compare PG-TD with, which also uses Transformer to evaluate partial programs. The pseudocode is shown in Alg. 3. One can think of it from a view of genetic algorithms. SMCG-TD maintains a group of partial programs, called a *generation*, and updates them iteratively. For each partial program in the generation, we sample the next token using the Transformer and append it to the partial program, and put it in a new generation. We then evaluate each partial program in the new generation in the same way as PG-TD: We use the Transformer to generate a complete program from the partial program and compute its reward (pass rate). We then use these rewards as the partial programs' fitness scores. At the end of an iteration, we re-sample from the partial programs according to their fitness. So partial programs with higher fitness scores are more likely to remain in the generation. In the end, the algorithm outputs the complete program it finds that has the largest reward.

One may notice that SMCG-TD resembles PG-TD on a higher level. It also uses the Transformer and the public test cases to decide which next token to generate in the code generation process. The key difference is that SMCG-TD uses a sampling-based approach and PG-TD uses a tree search algorithm. So in our experiments, the performance of SMCG-TD is worse than PG-TD.

It is worth noting that sequence caching is also implemented for SMCG-TD when it calls BEAM_SEARCH. So even if multiple partial programs in a generation can be the same, the algorithm would not call the Transformer to generate a complete program from the same prefix more than once, which effectively reduces the number of times that it needs to call the Transformer and saves the computation time.

### D.4    FINETUNING

In our experiments, we use the samples generated by the PG-TD to further finetune the Transformer model. We then generate solutions using the finetuned Transformer model by running beam search,

---

**Algorithm 3** Sequential Monte Carlo-Guided Transformer Decoding (SMCG-TD).

---

**Require:** $pop\_size$: the population size
1: $generation \leftarrow [\langle PD \rangle] * pop\_size$ # initialize the population with the problem description
2: $fitness \leftarrow [1.0/pop\_size] * pop\_size$ # uniform distribution
3: $t \leftarrow 0$
4: $complete\_programs \leftarrow$ NEW_LIST()
5: **while** $|population| > 0$ and $t < max\_steps$ **do**
6:     $new\_generation \leftarrow$ NEW_LIST()
7:     $new\_fitness \leftarrow$ NEW_LIST()
8:     **for** $i \leftarrow 1, \ldots,$ LEN($generation$) **do**
9:         $s \leftarrow$ sampled from $generation$, where $generation[j]$ is selected with the probability of $fitness[j]$
10:         $a \leftarrow$ sampled from $P_{Transformer}(\cdot|s)$
11:         $s' \leftarrow$ CONCAT($s, a$)
12:         **if** $s'[-1] = $ `<|endoftext|>` **then**
13:             # this sample finishes generation, save to output
14:             add $s'$ to $complete\_programs$
15:         **else**
16:             add $s'$ to $new\_generation$
17:             # BEAM_SEARCH uses sequence cache as in PG-TD
18:             $p \leftarrow$ BEAM_SEARCH($s', b = 1$)
19:             $r \leftarrow$ GET_REWARD($p$)
20:             add $r$ to $new\_fitness$
21:         **end if**
22:     **end for**
23:     normalize $new\_fitness$
24:     $generation \leftarrow new\_generation$
25:     $fitness \leftarrow new\_fitness$
26:     $t \leftarrow t + 1$
27: **end while**
28: **return** the program in $complete\_programs$ that has the largest reward

---

and observe improvement in both pass rate and strict accuracy compared with running beam search using the original model.

One challenge in our design is that solutions that PG-TD generates have different pass rates. We could use only solutions with a pass rate of 100%. However, that largely limits the number of samples we can use for finetuning. We instead collect all solutions that have pass rates larger than 80%. To make the Transformer aware of the different pass rates of training samples, we use the loss function similar to Le et al. (2022),

$$\mathcal{L}(\theta) = -\mathbb{E}_{W \sim p_\theta}[r(W)\nabla_\theta \log p_\theta(W|PD)], \tag{6}$$

where $W$ denotes a complete program; $r(W)$ is the pass rate of the program; $PD$ is the program description. In future work, we will investigate different loss functions and compare their effectiveness in improving the Transformer model's performance.

## E   ILLUSTRATIVE EXAMPLES

In this section, we provide more qualitative examples generated by our PG-TD and the baseline methods, shown in Figure 14 and Figure 15. Our PG-TD algorithm generates solutions that have higher pass rates than the baseline methods. We also show a concrete example where PG-TD outperforms beam search in Fig. 16 using Codex as the backbone Transformer.

For a step-by-step illustration of PG-TD, please visit our project website at https://codeaimcts.github.io.

---

*Problem Statement*

Takahashi is meeting up with Aoki. They have planned to meet at a place that is D meters away from Takahashi's house in T minutes from now. Takahashi will leave his house now and go straight to the place at a speed of S meters per minute. Will he arrive in time?

*Constraints*

(1). $1 \leq D \leq 10000$

(2). $1 \leq T \leq 10000$

(3). $1 \leq S \leq 10000$

(4). All values in input are integers.

*Input*

Input is given from Standard Input in the following format: $D\ T\ S$

*Output*

If Takahashi will reach the place in time, print $Yes$; otherwise, print $No$.

*Sample Test Input*

1000 15 80

*Sample Test Output*

Yes

It takes 12.5 minutes to go 1000 meters to the place at a speed of 80 meters per minute. They have planned to meet in 15 minutes so he will arrive in time.

---

```python
# cook your dish here
try:
    a=list(map(int,
    ↪   input().split(
    ↪   )))
    if(a[2]==1):
        print("Yes")
    else:
        print("No")
except:
    pass
```

Beam Search (Pass Rate: 0.125).

```python
# cook your dish here
d,t,s=list(map(int,
↪   input().split
↪   ()))
if(d==t):
    if(s*(s+1)/s==t):
        print("Yes")
    else:
        print("No")
else:
    if(s*(s+1)/s>t):
        print("Yes")
    else:
        print("No")
```

Sampling + Filtering (Pass Rate: 0.625).

```python
n, t, s=map(int,
↪   input().split())
t1=n / s
if t1 <= t:
    print("Yes")
else:
    print("No")
```

PG-TD (Pass Rate: 1.00).

Figure 14: A code generation example for competitive programming. The problem description is shown on the top. The programs generated by baseline algorithms and our PG-TD algorithm are shown at the bottom.

***Problem Statement***

*Given is an integer $r$. How many times is the area of a circle of radius $r$ larger than the area of a circle of radius $1$? It can be proved that the answer is always an integer under the constraints given.*

***Constraints***

*(1). $1 \le r \le 100$.*

*(2). All values in input are integers.*

***Input***

*Input is given from Standard Input in the following format: $r$.*

***Output***

*Print the area of a circle of radius $r$, divided by the area of a circle of radius $1$, as an integer..*

***Sample Test Input***

*2*

***Sample Test Output***

*4*

*The area of a circle of radius $2$ is $4$ times larger than the area of a circle of radius $1$. Note that output must be an integer - for example, $4.0$ will not be accepted.*

```
1  # cook your dish here
2  r=int(input())
3  if(r>=2*r):
4          print(r*r*2)
5  else:
6          print(0)
```

Beam Search (Pass Rate: 0.00).

```
1  # cook your dish here
2  r=int(input())
3  if(r>=2*r):
4          print(r*r*2)
5  else:
6          print(0)
```

Sampling + Filtering (Pass Rate: 0.00).

```
1  # cook your dish here
2  r=int( input() )
3  print(2*(r**2)-r**2)
4
```

PG-TD (Pass Rate: 1.00).

Figure 15: A code generation example for competitive programming. The problem description is shown on the top. The programs generated by baseline algorithms and our PG-TD algorithm are shown at the bottom.

---

**Problem Statement**

*You are given an array $a$ consisting of $n$ integer numbers.*

*Let instability of the array be the following value: $\max\limits_{i=1}^{n} a_i - \min\limits_{i=1}^{n} a_i$.*

*You have to remove exactly one element from this array to minimize instability of the resulting $(n-1)$-elements array. Your task is to calculate the minimum possible instability.*

**Input**

*The first line of the input contains one integer $n$ ($2 \le n \le 10^5$) — the number of elements in the array $a$.*

*The second line of the input contains $n$ integers $a_1, a_2, \ldots, a_n$ ($1 \le a_i \le 10^5$) — elements of the array $a$.*

**Output**

*Print one integer — the minimum possible instability of the array if you have to remove exactly one element from the array $a$.*

**Examples**

*Input*
*4*
*1 3 3 7*
*Output*
*2*
*Input*
*2*
*1 100000*
*Output*
*0*

**Note**

*In the first example you can remove 7 then instability of the remaining array will be $3 - 1 = 2$.*
*In the second example you can remove either 1 or 100000 then instability of the remaining array will be $100000 - 100000 = 0$ and $1 - 1 = 0$ correspondingly.*

---

```
1  n = int(input())
2  a = list(map(int, input().split()))
3
4  print(max(a) - min(a) - 1)
```

Beam search (Pass Rate 0.00).

```
1  def main():
2      n = int(input())
3      a = list(map(int, input().split()))
4      a.sort()
5      print(min(a[n-1] - a[1], a[n-2] - a[0]))
6
7  if __name__ == "__main__":
8      main()
```

PG-TD (Pass Rate 1.00).

Figure 16: A code generation example using Codex (code-davinci-002). Beam search finds an incorrect solution while PG-TD finds a correct solution. This result further confirms that PG-TD is model-agnostic and could be effective with different backbone Transformers (GPT-2 (1.5B), GPT-Neo (2.7B) and Codex).

## F    MORE DISCUSSIONS

**Potential benefits.**    We have shown that our framework can not only help achieve higher pass rates compared with using only the Transformer generation process, but also generate codes that optimize different objectives. These are done without fine-tuning the pre-trained Transformer models. Our framework makes a pre-trained Transformer more versatile and adapts it to different tasks by designing different reward functions used by the planning algorithm. If we re-train or fine-tune Transformers like GPT-2 used in this paper, we will need to train billions of parameters (Radford et al., 2019). Our approach potentially saves the computational cost and energy consumption that is needed to train or fine-tune Transformers for different tasks.

**Potential negative social impacts.**    Automatic code generation makes it easier for anyone to generate programs that meet a specification. Our hope is that this development will relieve the burden of software engineers and enable AI-assisted programming or even end-to-end automatic programming. However, it may also make it easier to develop malware. Anyone that can specify a malicious goal in a natural language can use an automatic code generation tool to generate codes that meet the goal. When automatic code generation techniques become more mature, we may need a separate module that screens the natural language description and rejects the code generation requests that can lead to harmful codes.

