# OpenReview forum: "Planning with Large Language Models for Code Generation"
_ICLR.cc/2023/Conference — ICLR 2023 poster_

### Official Review · Reviewer_FcyD · 2022-10-14

**Confidence:** 4
**Correctness:** 3
**Technical Novelty And Significance:** 3
**Empirical Novelty And Significance:** 3
**Recommendation:** 6

**Clarity, Quality, Novelty And Reproducibility:**

### Clarity
* The main parts of the paper are unclear and not explained:
    * I did not understand most of Section 3.
    * The main Figure 2 and Algorithm1 are barely explained, assuming that the reader is not only familiar but also proficient with Monte-Carlo Tree Search (MCTS).
    * For example, the procedure `UCB_SELECT` appears in the algorithm but not explained. I know that there are some additional details in the Appendix, but:
        1. `UCB_SELECT` is still not explicitly explained, and
        2.  in this case the appendices do not only provide "extra details", but are a crucial part of understanding the paper, and this makes the paper be not self-contained without reading the appendices.
* I did not understand Figures 2+3 (both because they were not explained, and because some of the text is tiny) and Algorithms 1+2.

### Questions to authors:
1. How do the authors see the proposed model to be useful in real life? In which settings and tasks?
2. "We assume that the agent has access to a set of test cases" - while this is the assumption of traditional "[programming by example](https://www.microsoft.com/en-us/research/wp-content/uploads/2016/12/pbe16.pdf)" approaches, in which modern realistic settings is this a reasonable assumption?
3. Figure 1 is not very motivating, because it shows an example that is quite degenerated (replacing a string `S` with `'x' * len(S)`), and claiming that standard beam search could not solve it.
Giving the same problem to Codex (in the OpenAI playground), using the same test input:
```
# Given a string, replace every character with 'x' and print the result.
# Example:
# replace_char('sardine')
# output: xxxxxxx

def replace_char(
```
which returned:
```
string):
    return 'x' * len(string)
```
Isn't this solution even better than PG-TD's solution in Figure 1? (except that PG-TD's solution makes an in-place modification while Codex's prediction returns a new string, but returning a new string is the more common, simpler, and more realistic solution anyways). I would like to see more motivating cases that show examples that the standard transformer could not solve at all, and were only solved thanks to PG-TD.

4. The paper claims that the standard approach of sampling many examples is not sample-efficient. However, isn't the exhaustive expansion in Figure 3 also not sample-efficient?

### Related work:
* How is the proposed approach different from Ellis et al. (2019) and Le et al. (2022)?
These works are mentioned in the related work, but I did not understand how do they differ from the proposed approach.

**Strength And Weaknesses:**

### Strengths
+ The problem is important and has a broad interest.
+ The idea is interesting conceptually and shows improved results.
+ The idea of treating code generation as an RL problem, where input-output pairs provide the reward, is novel and related to much work in program synthesis.

### Weaknesses
- **Usability** - I am finding it hard to see how the approach can be usable in realistic settings. This severe limitation is not explicitly mentioned, and is clear only to readers who are very familiar with the field and its datasets: while the paper places itself besides general models of code such as Codex,  CodeBERT, CodeT5 etc., the proposed approach is actually much more limited than them, and does not "play the same game". While Codex (for example) is only **tested** on executable test-cases to measure its execution accuracy, the proposed approach **requires training** on the input-output pairs. Further, not only that it requires input-output pairs at training time, at test time it can only be applied to examples that have input-output pairs as well.

In contrast, all other models (Codex,  CodeBERT, CodeT5 etc.) can be applied (at test time) to inputs that do not have input-output pairs, and this way they can serve as general code-completion / code-generation / bug-detection assistants.
The proposed approach kind of "overfits" on the existing benchmarks, while losing usability on the realistic tasks that these benchmarks are meant to be a proxy of.
How do the authors see the proposed model to be useful in real life? In which settings and tasks do we expect the user to provide both a natural language intent *and* test cases?

- **Clarity** - discussed in the next section.


**Summary Of The Paper:**

The paper proposes an approach for improving pretrained models of source code for generating better code, if the input includes test cases. Given a pretrained model of code, a natural language (NL) description of the problem, and a few input-output examples, the authors use a Monte-Carlo tree search (MCTS)-based algorithm to generate code that satisfies the input-output examples, instead of just sampling / using beam search and "hoping" that the generated code will satisfy the input-output examples (="pass the tests").
The idea is demonstrated on the APPS dataset, across GPT-2-1.5B and GPT-Neo-2.7B models, where the model passes more tests than the standard approaches of beam search or sampling + post-filtering.

**Summary Of The Review:**

While the idea is interesting conceptually, I think that the settings and assumptions are overly unrealistic, and I do not understand how can the proposed approach be ever useful in real life.

It would have improved the paper if the authors could show that a model that was **only trained** with test cases - works better at test time **without test cases at test time**. Otherwise, if the user of the model needs to provide both a NL intent *and* test cases for every test-example, isn't it a bit too much of a burden for the user of such a model?

I thus currently vote for rejection, but I am open to changing my mind if the authors convinced me otherwise.

=========================
## Post-discussion

The authors have resolved most of my concerns.
I think that the main weakness of the paper remains its limited usability, i.e., the setting it addresses is overall unrealistic.

I increased my score to 6, and I hope that the authors will complete the experiments where they generate the test cases as well, which will significantly strengthen the paper.

---

> ### Author Response · Authors · 2022-11-16
> **Response to Reviewer FcyD**
>
> Dear Reviewer FcyD,
>
> We thank the reviewer for the comments and questions.
>
> **Q1 and Q2. Rationale of using test cases.**
> > 1. How do the authors see the proposed model to be useful in real life? In which settings and tasks?
> > 2. "We assume that the agent has access to a set of test cases" - while this is the assumption of traditional "programming by example ..." approaches, in which modern realistic settings is this a reasonable assumption?
>
> We would like to emphasize that test cases are a unique feature that makes programming languages different from natural languages. Unlike natural language generation, where the goal is to generate sequences that resemble human languages, programming language generation tasks aim to generate programs that can successfully run and output the correct solutions. Even for one of the most capable code-generation Transformers, Codex, validating the generated programs by executing on a set of test cases can dramatically improve its performance (Table A). Therefore, **test cases are crucial to validate the correctness of the programs**. Nonetheless, it remains an open and challenging problem on how to make use of the provided test cases. Our paper aims to provide a novel decoding algorithm (PG-TD) to fill this gap.
>
> We agree with the reviewer that **"where the test cases come from"** is an important question that we need to answer to make our work more practically useful. There has been recent work on generating useful test cases for solving coding challenges. For example, AlphaCode [1] train an LLM-based test input generation model that generates test cases from natural language descriptions. They execute the generated programs on these test cases for clustering and filtering out low-quality programs. We evaluated our methods on the CodeContest dataset released in [1] （see table 1 of the main paper). In this dataset, **the majority of the test cases we used as public test cases are automatically generated**. This confirms that our framework works with automatically-generated test cases.
>
> Another real-world use case of our algorithm is code translation. The goal of code translation is to translate a script from one programming language to another. In this problem, automatically generated test cases are used to determine if the translated codes are correct [2]. The automatic test case generation problem is well-explored in this problem and **there exist tools like EvoSuite, Jtest, etc. that can generate useful test cases that can be used by PG-TD.** We conducted code translation experiments in the following table. We use TransCoder-ST [2] as the backbone Transformer, and compare beam search with PG-TD. In PG-TD, it uses EvoSuite to generate *public test cases*, and use the pass rates on them as rewards in the generation process. The translated codes are evaluated on a set of private test cases. We can see that in this problem, PG-TD also outperforms the beam search algorithm.
>
> Table E. Pass rates of applying our method for code translation, where test cases are automatically generated. The baseline results are copied from the original paper.
> |  | Python -> Java | CPP -> Java |
> | -------- | -------- | -------- |
> | Beam Search         | 58.2     | 68.0     |
> | PG-TD     | 63.5     | 70.6     |
>
> Therefore, **our PG-TD algorithm can be useful in multiple real-world coding tasks such as coding problem solving [1] and code translation [2].** We will add discussions on test cases and summarize related work to the final version of the paper.
>
> [1]. Yujia Li, David Choi, Junyoung Chung, Nate Kushman, Julian Schrittwieser, R ́emi Leblond, Tom Eccles, James Keeling, Felix Gimeno, and Agustin Dal Lago. Competition-Level Code Generation with AlphaCode. arXiv preprint arXiv:2203.07814, 2022.
>
> [2]. Baptiste Roziere, Jie M Zhang, Francois Charton, Mark Harman, Gabriel Synnaeve, and Guillaume Lample. Leveraging automated unit tests for unsupervised code translation. In International Conference on Learning Representations, 2022.
>
>
> **Q3. Qualitative and quantitative results using Codex.**
> > 3. I would like to see more motivating cases that show examples that the standard transformer could not solve at all, and were only solved thanks to PG-TD.
>
> We have added a new example where the Codex model fails to solve the problem using beam search, but successfully solves the problem (pass rate = 1) using PG-TD. Please see Fig. 15 in the revision.

---

> > ### Author Response · Authors · 2022-11-16
> > **Response to Reviewer FcyD (continued)**
> >
> > **Q4. Sample efficiency.**
> > > 4. The paper claims that the standard approach of sampling many examples is not sample-eìcient. However, isn't the exhaustive expansion in Figure 3 also not sample-eìcient?
> >
> > Thank you for your question on sample efficiency. By claiming that our algorithm is sample-efficient, we actually mean that our algorithm is more sample-efficient than the wide-used algorithm like the **Sampling+Filtering** algorithm. Besides, we have also developed novel caching techniques to further improve the MCTS's efficiency as shown in Figure 4 and Table 2.
> >
> > **Q5. Clarity and related work.**
> >
> > We thank the reviewer for the suggestions on making the paper more clear. We have added more descriptions of MCTS in the main paper and enlarging Fig. 2 to make it more readable. We also added the definition of `UCB_SELECT` after the definition of `UCB`. Hope these can address the reviewer's concerns.
> >
> > We have emphasized our contributions again in the general response. Regarding the related work, Ellis et al. (2019) use a sampling approach (Sequential Monte Carlo) for code generation. They consider a domain-specific language and does not use a large language model. We adapted their algorithm and used it as a baseline in our paper (SMCG-TD). Le et al. (2022) similarly use a reinforcement learning formulation. However, they use a reinforcement learning objective to *train* a code-generaion Transformer. We instead use the RL formulation in the decoding process of a pre-trained code-generation Transformer.

---

> > ### Comment · Reviewer_FcyD · 2022-11-18
> > **Response**
> >
> > >Even for one of the most capable code-generation Transformers, Codex, validating the generated programs by executing on a set of test cases can dramatically improve its performance (Table A). Therefore, **test cases are crucial to validate the correctness of the programs**
> >
> > This does not answer my question. Of course, test cases are crucial. But as I asked, how do the authors expect the proposed model  to be useful in real life?
> >
> >
> > > In this dataset, the majority of the test cases we used as public test cases are automatically generated. This confirms that our framework works with automatically-generated test cases.
> >
> > I did not read the CodeContest dataset paper, but I am assuming that these test cases were generated **after** the code existed?
> > So why is it meaningful to generate code given these test cases?
> > Otherwise, how could the test cases be generated before the code was written?
> >
> > In other words, it seems like the authors claim that "test case generation is a solved problem".
> > I don't think this is true, and it requires more empirical evidence and end-to-end evaluation.

---

> > > ### Author Response · Authors · 2022-11-19
> > > **Response to Reviewer FcyD**
> > >
> > > We thank the reviewer for the follow-up questions and insightful feedback.
> > >
> > > > I did not read the CodeContest dataset paper, but I am assuming that these test cases were generated after the code existed? So why is it meaningful to generate code given these test cases? Otherwise, how could the test cases be generated before the code was written?
> > > > In other words, it seems like the authors claim that "test case generation is a solved problem". I don't think this is true, and it requires more empirical evidence and end-to-end evaluation.
> > >
> > > We could use a machine learning model to generate not only *codes*, but also *test cases* from natural language descriptions. Indeed, the AlphaCode paper [1] has shown promising results. In their CodeContests dataset, **the test case inputs are automatically generated from the natural language descriptions of the problems**. As they have stated in the paper,  *"We trained a separate test input generation model, using the same architecture as our main models, and initialised from the same GitHub pre-trained checkpoint. **This model was trained to predict test inputs from problem descriptions**, using example, hidden, and generated test inputs as training data. After training, this model was used to create new test inputs for unseen problems."* (Page 13 of Section 4.6 Clustering in [1])
> > >
> > > It would be indeed more challenging to automatically generates *both the input and output* of test cases. However, if we use a neural network that generates test cases used by our algorithm, we could use a co-training or GAN-like training diagram to improve both the performance of our code generation algorithm and the quality of the test cases. We will leave it for future work.
> > >
> > > We'd also like to clarify that we do **not** believe that test case generation is a solved problem. As we have reviewed in the related work section, **many works in the code generation literature rely on test cases for code generation [1-5].** We agree with the reviewer that it remains an important question how to automatically generate useful test cases (input, output pairs) to help algorithms that use test cases like ours.
> > >
> > > > Of course, test cases are crucial. But as I asked, how do the authors expect the proposed model to be useful in real life?
> > >
> > > Our proposed model is useful in the following ways.
> > >
> > > First, we can **train a separate model to predict test cases from natural language problem descriptions**. We have shown empirical evidence that when evaluated on CodeContests (in Table 1), our algorithm shows significant improvement over beam search using test cases where the test inputs are automatically generated.
> > >
> > > Second, we want to emphasize that **our method does not require many test cases**. We have shown in Table 7 in the appendix that even with one or two public test cases, our method shows significant improvement over the beam search baseline. If a human user is willing to specify *one or two* test cases, that can significantly help our algorithm. Although, as we expect, more test cases can further help our algorithm.
> > >
> > > Lastly, **our algorithm is versatile and can be used for other code generation problems**. In Table E, we show that our algorithm can be applied to a code translation task and has a better performance than the beam search baseline. In this problem, the code in the source language already exists so it is easier to automatically generate test cases.
> > >
> > > To summarize, although the important problem of automatic test generation is beyond the scope of our paper, we could benefit from automatically-generated test cases like in [1] or use a small number of human-designed test cases if available. Please don’t hesitate to let us know if there are any additional clarifications that we can offer. We would love to convince you of the merits of the paper. We appreciate your suggestions. Thanks!
> > >
> > >
> > > [1]. Yujia Li, David Choi, Junyoung Chung, Nate Kushman, Julian Schrittwieser, R ́emi Leblond, Tom Eccles, James Keeling, Felix Gimeno, and Agustin Dal Lago. Competition-Level Code Generation with AlphaCode. arXiv preprint arXiv:2203.07814, 2022.
> > >
> > > [2]. Bunel, Rudy, et al. "Leveraging grammar and reinforcement learning for neural program synthesis." arXiv preprint arXiv:1805.04276 (2018).
> > >
> > > [3]. Chen, Xinyun, Chang Liu, and Dawn Song. "Execution-guided neural program synthesis." International Conference on Learning Representations. 2018.
> > >
> > > [4]. Chen, Xinyun, Dawn Song, and Yuandong Tian. "Latent execution for neural program synthesis beyond domain-specific languages." Advances in Neural Information Processing Systems 34 (2021): 22196-22208.
> > >
> > > [5]. Xu, Yifan, et al. "Neural Program Synthesis By Self-Learning." arXiv preprint arXiv:1910.05865 (2019).

---

> > > ### Author Response · Authors · 2022-11-23
> > > **Looking forward to more discussion**
> > >
> > > Dear Reviewer FcyD,
> > >
> > > We would like to thank you again for your feedback and questions. If you have any other concerns, we would like to continue the discussion!
> > >
> > > We would like to mention the concurrent work [1] that does **generate test cases from natural language descriptions**. The work is currently under review. However, it confirms that large language models like Codex can generate useful test cases as well. As shown in the example in Fig. 2 of the paper [1], With a function signature, comments on the function, and an `assert` in the prompt, the Codex model can generate a list of assertion as test cases. We could exploit this approach to automatically generate test cases and use them in our framework, which makes our algorithm more practical in the real world. We leave this for future work.
> > >
> > > [1] *Anonymous authors*. CodeT: Code Generation with Generated Tests. https://openreview.net/pdf?id=ktrw68Cmu9c

---

> > > > ### Comment · Reviewer_FcyD · 2022-11-23
> > > > **I don't think this is allowed**
> > > >
> > > > I don't think that continuing the discussion is allowed.
> > > > The discussion period has ended on November 18.
> > > >
> > > > >Codex model can generate a list of assertion as test cases
> > > >
> > > > Codex *can* generate test cases. But are these assertions correct? are they useful? are they accurate enough to rely on them in further generation of code?
> > > >
> > > > >We could exploit this approach to automatically generate test cases and use them in our framework, which makes our algorithm more practical in the real world
> > > >
> > > > You **could**, but currently, the paper does not include such results.
> > > >
> > > > I would have supported this paper if it showed the ability to generate test cases and then condition on them, end-to-end. However, all these claims that
> > > > * "The automatic test case generation problem is well-explored in this problem and there exist tools",
> > > > * "we can train a separate model to predict test cases from natural language problem descriptions",
> > > > * "Codex can generate useful test cases"
> > > >
> > > > require further empirical evidence, and especially evidence that the generated tests are useful for conditioning on them in a downstream task, and thus did not convince me that the proposed approach is useful beyond unrealistic benchmarks

---

> > > > > ### Author Response · Authors · 2022-12-03
> > > > > **Response to Reviewer FcyD**
> > > > >
> > > > > We would like to emphasize that our PG-TD algorithm can work with either human-specified test cases or automatically-generated ones (which we will show in this response). If a human user is willing to specify even a small number of test cases, our algorithm can perform dramatically better than beam search (Table 7 in the paper).
> > > > >
> > > > > When human-specified test cases are not available, our algorithm can still rely on automatically-generated test cases. We agree with the reviewer that it is an important and challenging problem to generate useful test cases for code generation. In this response, we provide additional experimental results to answer the following questions.
> > > > > 1. *Can large language models (like Codex) generate correct test cases?*
> > > > > 2. *Can the PG-TD algorithm take advantage of the automatically-generated test cases to generate better programs?*
> > > > >
> > > > > We first follow [1] to use the Codex model and evaluate the code generation algorithms on the HumanEval dataset [2]. We adopt the same procedure as [1] to automatically generate test cases. Concretely, we use a prompt that includes the natural language description of the problem and an `assert` statement at the end of the prompt. We also **removed the example input, output pairs in the problem description** to avoid the Transformer from directly copying the test cases in the problem description. The following is an example of the prompt for test case generation.
> > > > > ```
> > > > > from typing import List
> > > > >
> > > > > def has_close_elements(numbers: List[float], threshold: float) -> bool:
> > > > >     """ Check if in given list of numbers, are any two numbers closer to each other than
> > > > >     given threshold.
> > > > >     """
> > > > >     pass
> > > > >
> > > > > # check the correctness of has_close_elements
> > > > > def check(candidate):
> > > > >     assert candidate
> > > > > ```
> > > > >
> > > > > **Does Codex generate correct test cases?** In the HumanEval dataset, the generated solutions are evaluated by the test script by calling the `check` function. So we use the strict accuracy (pass@1) metric that counts the percentage of the problems where the generated solutions pass the `check` function.
> > > > >
> > > > > To evaluate the quality of the automatically-generated test cases, we compute the strict accuracies of the *sample solutions* (correct solutions written by human programmers) on them. The strict accuracies of the sample solutions on the *ground-truth test cases* should be 100%. We **evaluate the sample solutions on the automatically-generated test cases and find the corresponding strict accuracy is 72.56%** over the whole HumanEval dataset, which confirms that the automatically-generated test cases are mostly correct.
> > > > >
> > > > > **Can PG-TD take advantage of the automatically-generated test cases?** Since OpenAI restricts the frequency of using the Codex model, we have been striving to run the PG-TD algorithm using the automatically-generated test cases to verify the test cases' effectiveness. We report the average strict accuracy of the generated programs on a subset of HumanEval problems in Table H. We can confirm that the strict accuracy is higher for PG-TD as it uses high-quality automatically-generated test cases to verify the generated programs. We are currently running experiments on more problems to make the results more comprehensive.
> > > > >
> > > > > We also find the pattern that using beam search, the Codex model tends to generate redundant codes, which eventually make the program fail to execute. This issue is resolved by PG-TD, which verifies the generated programs using the automatically-generated test cases.
> > > > >
> > > > > Table H. Comparison between beam search and TG-PD using automatically-generated test cases, using the Codex model and evaluated on the first 20 problems in the HumanEval dataset.
> > > > > |             | Strict Accuracy (%) |
> > > > > | ----------- | --------- |
> > > > > | Beam Search | 30.0      |
> > > > > | PG-TD (using auto-generated test cases) | 50.0          |
> > > > >
> > > > >
> > > > > [1] *Anonymous authors*. CodeT: Code Generation with Generated Tests
> > > > > https://openreview.net/pdf?id=ktrw68Cmu9c
> > > > >
> > > > > [2] Chen, Mark, et al. "Evaluating large language models trained on code." arXiv preprint arXiv:2107.03374 (2021).
> > > > >
> > > > > We wish that our response has addressed your concerns, and turns your assessment to the positive side. If you have any questions, please feel free to let us know during the rebuttal window. We appreciate your suggestions and comments! Thank you!
> > > > >
> > > > > Best,
> > > > >
> > > > > Authors

---

> > > > > > ### Comment · Reviewer_FcyD · 2022-12-03
> > > > > > **Disregarding these results**
> > > > > >
> > > > > > Dear authors,
> > > > > >
> > > > > > As I wrote in the previous message -
> > > > > > the discussion period had ended 14 days ago. I cannot regard these new results, as it will be unfair to the rest of the authors.

---

> > > > > > > ### Author Response · Authors · 2022-12-03
> > > > > > > **Response to Reviewer FcyD**
> > > > > > >
> > > > > > > Dear Reviewer,
> > > > > > >
> > > > > > > We thank you for getting back to us and expressing your concern on whether discussions at this phase follow the ICLR guideline.
> > > > > > >
> > > > > > > We have carefully reviewed the ICLR 2023 author guideline. The authors, reviewers, and AC are allowed to have discussions at this stage. We also find some other papers are still under active discussions. The following is quoted from the ICLR 2023 author guideline.
> > > > > > > > * Discussion Stage 2 - November 18 - December 12
> > > > > > > >     * **Discussions among authors, reviewers and AC (authors can’t update their submissions during this time)**
> > > > > > > >     * [new this year] ACs will meet with reviewers for borderline papers
> > > > > > > >     * ACs encourage reviewers to acknowledge and respond to author responses. [1]
> > > > > > >
> > > > > > > [1] https://iclr.cc/Conferences/2023/AuthorGuide#:~:text=Discussions%20among%20authors%2C%20reviewers%20and%20AC%20(authors%20can%E2%80%99t%20update%20their%20submissions%20during%20this%20time)
> > > > > > >
> > > > > > > We sincerely appreciate the feedback the reviewer has provided so far. As discussions are indeed allowed, we hope to take advantage of this phase to address the reviewer's misunderstanding, better understand our algorithm, and make our paper stronger. Our goal is to make scientific advances to solve the code generation problem. This is possible only if we, other reviewers and AC are having a thorough discussion on the paper.
> > > > > > >
> > > > > > > Best,
> > > > > > >
> > > > > > > Authors

---

> > > > > > > ### Author Response · Authors · 2022-12-06
> > > > > > > **Regarding policy during discussion phase**
> > > > > > >
> > > > > > > Dear Reviewer,
> > > > > > >
> > > > > > > We have contacted the program chairs about the policy during the discussion phase. Discussions between the reviewers and authors are indeed allowed in this phase. Here's the email from the program chair.
> > > > > > >
> > > > > > > > Hi [Author],
> > > > > > > > This period is mainly for reviewers and AC to discuss, but authors can still post msgs and answer questions. But you can refer to us that discussion between authors and reviewers is still allowed.
> > > > > > > > Mengdi
> > > > > > >
> > > > > > > As the deadline for the discussions is approaching, we hope to address any concerns that the reviewer may have. Please don't hesitate to let us know if you have any questions or concerns regarding our response.
> > > > > > >
> > > > > > > Thanks,
> > > > > > >
> > > > > > > Authors

---

> > > > > > ### Comment · Reviewer_FcyD · 2022-12-08
> > > > > > **I increased my score to 6**
> > > > > >
> > > > > > The authors have resolved most of my concerns. I think that the main weakness of the paper remains its limited usability, i.e., the setting it addresses is overall unrealistic.
> > > > > >
> > > > > > I increased my score to 6, and I hope that the authors will complete the experiments where they generate the test cases as well, which will significantly strengthen the paper.

---

> > > > > > > ### Author Response · Authors · 2022-12-12
> > > > > > > **Thank you**
> > > > > > >
> > > > > > > Dear Reviewer FcyD,
> > > > > > >
> > > > > > > Thank you for reviewing our new results and we are glad that they have resolved your concerns. We also thank you for recognizing the merits of the paper and raising the score. We agree that "where the test cases come from" is an important question. We will include the complete results on using automatically-generated test cases in the main paper. Thanks again for your insightful suggestions to strengthen our work.
> > > > > > >
> > > > > > > Thanks,
> > > > > > >
> > > > > > > Authors

---

### Official Review · Reviewer_kmKW · 2022-10-16

**Confidence:** 4
**Correctness:** 3
**Technical Novelty And Significance:** 2
**Empirical Novelty And Significance:** 3
**Recommendation:** 8

**Clarity, Quality, Novelty And Reproducibility:**

### Clarity
The paper is fairly clearly written. However, the fact that the authors refer to the method as "MCTS" despite not having any sampling steps is a bit misleading.

One minor comment: on page 2, I'm not sure what the authors mean by "by treating programming languages as particular languages." What is a "particular language" in this context?

### Quality
As mentioned in the weaknesses section, I think that the proposed algorithm is not really MCTS due to not having any stochastic (Monte Carlo) sample evaluation procedure. Using truly random rollouts would yield a more faithful version of MCTS, and give a better signal of whether MCTS is useful for LLM program generation. (At the very least, it would be good to discuss this difference more explicitly in the paper, and perhaps only refer to the proposed method as being inspired by MCTS.)

I also have a number of concerns with the quality of the experimental results. It's not clear to me how the hyperparameters for their approach or for the baselines were selected, and some choices seem unfair. For instance, their method only considers the top 3 tokens at each step, whereas sample-and-filter uses top 5 sampling, and SMCG-TD may not use any top-k sampling. This means that their method is exploring a significantly smaller search space than all of the baselines! So if good programs exist under a top-3 restriction, their method has a much higher chance of finding them. It seems important to use the same search space for all of the baselines (i.e. use the same top-k restriction) in order to fairly compare the node-expansion strategies.

It's also hard for me to reconcile the results in Table 1 with those in Figure 4. In Table 1, PG-TD appears to often outperform SMCG-TD by a large margin. However, in Figure 4, the two methods have very similar performance at a fixed number of generated programs or at a fixed runtime. Since the hyperparameters directly affect both the number of generated programs and the accuracy of each method, it seems critical that the comparisons in Table 1 be conducted for some fixed value of these. For instance, one could select the number of generated programs for sample-and-filter and for SMCG-TD to match the number of generated programs explored by their method.

### Novelty
I have not seen any other works that uses both MCTS and large language models to do program generation. However, there are previous works that do pairwise combinations of these (e.g. MCTS with a language model, MCTS for program generation, language models for program generation), so the combination of all three is only marginally novel technically speaking.

The two caching methods proposed by the authors seem interesting and useful, although they appear to be fairly straightforward applications of memo-ization techniques rather than being totally novel ideas.

Regardless of the technical novelty, there is value in providing new empirical evidence that these techniques work well in the problem synthesis setting. However, given my concerns about the experimental results, I'm not sure the evidence is strong enough to be significant.

### Reproducibility
The paper describes their approach clearly enough that I think it could be reimplemented, although it is fairly complex so this might require a bit of effort. I think the experiments could be reimplemented as well based on the information  provided, but I'm not particularly confident the findings would be robust to slightly different hyperparameter choices or implementations.

**Strength And Weaknesses:**

Strengths:

- Integrating LM generation with MCTS is a good idea, allowing the search process to focus on prefixes with a higher chance of producing good programs.
- The ability of their approach to cache and reuse outputs from the beam-search rollouts means the proposed strategy can be implemented fairly efficiently.
- Their method seems to slightly outperform baseline methods given a fixed runtime budget, without needing to retrain the transformer model.

Weaknesses:

- **(mostly resolved in new revision)** Although the authors refer to their planning algorithm as MCTS, the "Monte Carlo" aspect appears to be missing. The proposed approach is deterministic: it selects actions based on the most-likely tokens following a prefix, and computes "rollouts" using either beam search or greedy decoding.
  - The algorithm thus seems more like a heuristic best-first search than MCTS. Since a lot of the value of MCTS comes from aggregating over randomized rollouts, I would imagine randomization might lead to better performance. (On the other hand, the proposed "sequence caching" wouldn't work for random rollouts.)
  - This choice is also confusing from a clarity standpoint; I expected the rollouts to be sample-based given that the method was based on MCTS.
  - *The authors have rephrased this to not say their method is exactly MCTS, although I do have some remaining suggestions for how to make this clearer; see discussion.*
- The main evaluation results are based on a seemingly arbitrary set of hyperparameters for each decoding strategy, and it's not clear to me that the comparisons (e.g. in Table 1) are fair.
  - **(resolved in new revision)** As one example, their method explores only the top 3 tokens at each state, but their sample-and-filter baseline explores the top 5, and SMCG-TD doesn't have any top-k restriction (at least not stated in the paper).
  - **(resolved in comments)** Additionally, the plots in Figure 4 show much smaller differences between their method and SMCG-TD than the differences in Table 1. This suggests that the differences in Table 1 are partially explained by generating different numbers of programs, or running for different amounts of time, due to the arbitrary hyperparameters selected for comparison.
    - *The authors have included updated results in the comments below. Indeed their approach generates more programs by default. For a small budget, their approach is slightly worse than the SMCG-TD baseline, whereas for a large budget their approach is better. Their approach is also faster relative to the number of unique programs it generates.*
- The novelty of the approach seems fairly small relative to ordinary MCTS and to existing applications of it to LLMs (e.g. the PPL-MCTS approach of Chaffin et al. (2022)).

**Summary Of The Paper:**

This paper proposes using a (MCTS-inspired) heuristic planning algorithm to guide the decoding process of a pre-trained code-generation transformer, for the specific task of test-case-guided competitive programming. At a high level, the proposed decoding strategy maintains a tree of prefixes, conducts beam-search-based "rollouts" and evaluates them on test cases to assess quality of these prefixes, then uses this information to determine which tree nodes to continue exploring. (I believe that the proposed method is deterministic, in contrast to MCTS which is stochastic by design.)

Since beam search and greedy decoding already involve expanding a tree of tokens, the authors introduce caching mechanisms to make their proposed planner more efficient: they cache the tree structure generated during beam search and re-use it when expanding nodes for MCTS, and (when using greedy rollouts) they cache the greedy decoding output for any given prefix and re-use it when expanding the nodes along an already-explored trajectory.

The authors compare their approach to beam search, sample-and-filter approaches (such as the one used by AlphaCode), and the SMCG-TD decoding strategy proposed by Ellis et al. (2019). They argue that their decoding strategy outperforms these other strategies, although some of this may be due to the particular hyperparameters they used; for a fixed number of generated programs or fixed computation time their approach seems to perform fairly similarly to SMCG-TD.

**Summary Of The Review:**

Although combining MCTS with a LLM for program generation is a good idea, there is not much technical novelty, and I have concerns about the validity of the empirical claims and whether it is appropriate to call the proposed approach MCTS given that it removes the stochastic part. I don't think this paper currently meets the bar for acceptance.

I think this paper could be strengthened by
- investigating the behavior of the planning algorithm when rollouts are actually drawn randomly rather than via greedy search or beam search (in other words, when the planning algorithm is actually MCTS)
- being more careful about hyperparameter choices for the baselines to ensure any differences are due to the differences in action selection / exploration / planning, rather than being due to differences in search space size or generated program budgets.

**Updated review:** The authors have addressed my concerns: they have clarified the relationship to MCTS, adjusted baselines to use more similar hyperparameters, and ran a comparison of beam search to sampling-based MCTS (for which their beam search method does better). In the discussion thread below, they have also included additional comparisons for Table 1 (labeled Table F in the comments below) that give a better sense of the performance of their approach; it consistently outperforms sampling-and-filtering, and given either a fixed computation time or a large enough budget, it also seems to outperform the SMCG-TD baseline (although only by a small margin). I have updated my score from 3 to 8, with the understanding that the updated results will be included in the final version of the paper if accepted.

---

> ### Author Response · Authors · 2022-11-16
> **Response to Reviewer kmKW**
>
> Dear Reviewer kmKW,
>
> We thank the reviewer for the detailed reviews and insightful suggestions.
>
> **Q1. Sampling-based MCTS.**
> > Although the authors refer to their planning algorithm as MCTS, the "Monte Carlo" aspect appears to be missing.
> > At the very least, it would be good to discuss this difference more explicitly in the paper, and perhaps only refer to the proposed method as being inspired by MCTS.
>
> We could use a sampling approach in the evaluation step of MCTS instead of using beam search, which would be a more faithful implementation of MCTS. We experimented with the two versions of PG-TD and found that using beam search indeed has a better performance. The gap could be caused by that we only do sampling once to evaluate a new node. We could get a more accurate value estimation by sampling multiple times. However, that would make our algorithm more computationally expensive.
>
> We also agree with the reviewer that it could be misleading to say our PG-TD algorithm uses MCTS. We have modified the method section to say "we consider a tree search-based planning algorithm *inspired by Monte-Carlo tree search (MCTS)*."
>
> Table B. PG-TD using beam search vs. sampling in the evaluation step, evaluated on the first 500 APPS introductory problems.
> | Decoding Algorithm     | Pass Rate | Strict Accuracy |
> | ---------------------- | --------- | --------------- |
> | PG-TD using Beam Search (as in paper) | 32.757          | 13.20                |
> | PG-TD using Sampling | 15.69          | 5.0                |
>
>
> **Q2. Consistent baseline hyper-parameters.**
> > The main evaluation results are based on a seemingly arbitrary set of hyperparameters for each decoding strategy, and it's not clear to me that the comparisons (e.g. in Table 1) are fair.
>
> We thank the reviewer for the suggestion and have rerun the experiments with the same hyperparamters.
> As stated in the general response to all reviewers above, we used top-k sampling where k=5 in our original submission, hoping this benefits sampling. For other experiments, we use k=3 (except when we explicitly change the tree width in the variations of PG-TD in Fig. 4). We now set k=3 for both our algorithms and the baseline algorithms, that is, only the first three most-likely tokens are considered in each generation step. We have updated Fig. 4 to reflect the change.
>
> > Since the hyperparameters directly affect both the number of generated programs and the accuracy of each method, it seems critical that the comparisons in Table 1 be conducted for some fixed value of these. For instance, one could select the number of generated programs for sample-and-filter and for SMCG-TD to match the number of generated programs explored by their method.
>
> We thank the reviewer for the suggestion. This would make the comparison in Table 1 more fair and consistent with Fig. 4. We will also update Table 1 to report the results under the same number of generated programs.
>
> **Q3. Result explanations.**
> > the plots in Figure 4 show much smaller differences between their method and SMCG-TD than the differences in Table 1. This suggests that the differences in Table 1 are partially explained by generating different numbers of programs, or running for different amounts of time, due to the arbitrary hyperparameters selected for comparison.
>
> We are sorry about the confusion on the "inconsistent" numbers between Table 1 and Fig. 4. The experiments in Table 1 are conducted on the whole APPS introductory dataset (**1000** problems), while for our more detailed analysis in Fig. 4, we only use the first **500** problems of the APPS introductory dataset. We are running the experiments of all the algorithms in Fig. 4 and will update the figures so that it is evaluated on the same dataset as in Table 1 to avoid confusion.
>
> We have explained in the *Efficiency of PG-TD* paragraph that SMCG-TD is close to PG-TD when PG-TD does one-step look-ahead planning (for example, when $width=rollouts=3$). When we increase the beam size ($b$) or the number of rollouts ($rollouts$), PG-TD has a clearly better performance than SMCG-TD (Fig. 4).

---

> > ### Author Response · Authors · 2022-11-16
> > **Response to Reviewer kmKW (continued)**
> >
> > **Q4. Related work on using MCTS with LLMs.**
> > > The novelty of the approach seems fairly small relative to ordinary MCTS and to existing applications of it to LLMs (e.g. the PPL-MCTS approach of Chaìn et al. (2022)).
> >
> > We have discussed other works that use MCTS and LLMs in *Planning in natural language processing* in the related work. As we have stated in the section, all these previous works use a pretrained neural network or an existing metric as a reward function. Take PPL-MCTS for example, they combine MCTS with a pre-trained discriminator instead of an LLM. The discriminator can be used to evaluate any states (partial or complete ones), which can be easily integrated with MCTS to estimate the value of a node (or state). However, in our case, the code generation LLM can only evaluate complete programs and is more expensive to run. We'll need to generate complete programs using beam search and design caching mechanisms to reduce repetitive generation calls to the LLM.
> >
> > We want to again emphasize the following contributions.
> > 1. We are the first to combine MCTS with large language models for general-purpose programming language generation. Our algorithm achieves significantly better performance than benchmarking algorithms in the literature.
> > 2. We have designed the interfaces between these two components and dealt with the unique challenges of making the framework computationally efficient.
> > 3. We have also demonstrated the effectiveness of the proposed PG-TD for other coding targets like generating shorter codes and codes with more comments in the experiment section, and show that codes generated PG-TD can better finetune the LLM.
> >
> > As the reviewer has noted, there are works that integrate MCTS with LLM for *natural language processing* tasks, which we have also discussed in our related work section. However, it remains an open and challenging problem how to effectively develop an algorithm that handles the unique challenges of code generation. For example, how to make use of the available test cases to develop a better code generation decoder. How to design an algorithm that avoids repeated computation during the search process.
> >
> > **Q5. Clarification questions.**
> > > One minor comment: on page 2, I'm not sure what the authors mean by "by treating programming languages as particular languages." What is a "particular language" in this context?
> >
> > We are sorry for the confusion. We meant "the Transformer treats the programming languages in the same way as natural languages." Since transformer-based language models have shown impressive capacities on natural language tasks, we can use similar transformer-based models to handle programming languages by treating programming language data as particular natural language data. We have modified the section to improve clarity.

---

> > > ### Comment · Reviewer_kmKW · 2022-11-21
> > > **Continued discussion and updated review**
> > >
> > >
> > >
> > > **Q1. Samping-based MCTS**
> > >
> > > Thank you for running the experiment comparing sampling to beam search. It is interesting that the beam search performs better than sampling here (and I wonder how many samples would be necessary for sampling to reach the performance of beam search) but this seems like a question out of scope for this work.
> > >
> > > Thanks also for rephrasing the paper to say it is inspired by MCTS (although I do still notice that parts of 3.2 and 3.3 still refer to the algorithm as MCTS directly). In the next revision I would also suggest adding a sentence in the last paragraph of section 3.2 that explicitly mentions that doing beam search here is different from the traditional random sampling of MCTS, since I still think the differences here are potentially confusing for readers.
> > >
> > > **Q2. Consistent baseline hyper-parameters.**
> > >
> > > Thanks for changing the baselines to use the same top-k restriction as the proposed method.
> > >
> > > It looks like Table 1 has not yet been updated to report results for the same number of generated programs; are these experiments still running?
> > >
> > > **Q3. Result explanations.**
> > >
> > > Sorry for being unclear in my initial review, I understand that the numbers won't be exactly the same because they are on different datasets. However, I would still expect the relative ranking of methods and differences between them to be similar on the first 500 problems and the full set of 1000 problems. So my concern was that Table 1 seems to show a big difference between PG-TD and SMCG-TD, whereas Figure 4 seems to show that PG-TD performs very similarly to SMCG-TD for a fixed number of generated programs, which made me worry that the comparison in Table 1 might be unfair (e.g. it might use more generated programs for PG-TD). I think my concern here will be addressed by updating Table 1 to use a fixed number of generated programs for each method, as mentioned for Q2 above, but I would worry that this updated Table 1 will not show much of an improvement from SMCG-TD to PG-TD, since it is plausible to me that the higher accuracy of PG-TD is mostly due to generating more programs (regardless of which dataset it is evaluated on).
> > >
> > > On this note, I don't agree that Figure 4 provides evidence that increasing $b$ or the number of rollouts leads to "clearly better performance" than SMCG-TD. From Figure 4, I would say that increasing $b$ or the number of rollouts leads to a larger number of generated programs, and that a larger number of generated programs leads to improved performance. However, SMCG-TD was never evaluated with hyperparameters that allow it to generate 1500 programs. So there is no way to tell from Figure 4 whether SMCG-TD would *also* have this high of performance if you scaled it up to generate 1500 programs. Indeed, from the left side of Figure 4, it looks to me like SMCG-TD and all variants of PG-TD are roughly following the same curve of performance relative to number of generated programs. (There is perhaps still an advantage in terms of computation time, but again, SMCG-TD was never run for 2000 sec either, so the comparison requires a bit of extrapolation.)
> > >
> > > **Updated review**
> > >
> > > I have increased my score from 3 to 5 in recognition of the wording changes in the paper, the additional experiments with sampling and with fixed $k=3$, and the stated plan to change Table 1 to compare with a fixed number of generated programs. However, since the updates to Table 1 have not yet been added, I'm not yet confident that the paper's claims (e.g. "our algorithm can consistently generate better programs in terms of the pass rate and other metrics compared with the baseline methods") are well supported, and that the benefits of the proposed approach are not just artifacts from generating more programs.

---

> > > > ### Author Response · Authors · 2022-11-23
> > > > **Response to Reviewer kmKW**
> > > >
> > > > We thank the reviewer for reviewing our new results and the follow-up questions.
> > > > We will revise the paper to make sure we mention it as a tree search algorithm, and also emphasize that we use beam search in the evaluation step of our tree search algorithm.
> > > >
> > > > Regarding Table 1, we have some updated results below on the pass rates of different algorithms **under the same number of generated programs** (Table F). In this table, we use the APPS GPT-2 (1.5B) model as the backbone Transformer and evaluate our algorithms on the APPS introductory dataset. We essentially limit the number of function calls to the Transformer generation function so the generated programs are limited. For PG-TD and SMCG-TD, we make them stop early when the program generation budget is reached and return the best solution found so far. We still fix $k=3$, that is, only the top-3 most-likely tokens are considered for all algorithms. We report the results of PG-TD under different rollout times and beam sizes (b), and SMCG-TD under different population sizes.
> > > >
> > > > Based on the results in Table F below, we have the following observations. First, both PG-TD and SMCG-TD outperform the Sampling + Filtering method. This confirms the benefit of integrating a planning or search algorithm into the Transformer beam search algorithm for code generation. Second, PG-TD finds the best program with a larger program generation budget (when the budget is 1024 or $\infty$). Particularly, PG-TD (rollout=9, b=5) has the best performance when no program generation budget is imposed. This shows our algorithm's effectiveness in searching and finding better solutions. We also notice that with a smaller number of generated programs (i.e. 512), SMCG-TD (population=100) achieves slightly better performance than our algorithm. We believe the reason is that a small program generation budget makes the tree search terminate too early. It may not have carefully searched the lower (deeper) part of the tree.
> > > >
> > > >
> > > > Table F. Pass rates (%) of PG-TD and other algorithms under different program generation budgets.
> > > > | Algorithm \ Program Generation Budget | 512       | 1024      | $\infty$  |
> > > > | ------------------------------ | --------- | --------- | --------- |
> > > > | PG-TD (rollout=3, b=1)         | 23.11     | 23.14     | 23.14     |
> > > > | PG-TD (rollout=3, b=3)         | 24.16     | 24.45     | 26.53     |
> > > > | PG-TD (rollout=6, b=3)         | 24.42     | 24.73     | 26.93     |
> > > > | PG-TD (rollout=9, b=3)         | 24.6      | **25.92** | 27.8      |
> > > > | PG-TD (rollout=9, b=5)         | 22.85     | 24.24     | **27.68** |
> > > > | SMCG-TD (population=50)        | 23.76     | 23.76     | 23.76     |
> > > > | SMCG-TD (population=100)       | **25.02** | 25.04     | 25.04     |
> > > > | Sampling + Filtering           | 16.75     | \-        | \-        |
> > > >
> > > > | Algorithm \ Program Generation Budget | 5|
> > > > | ------------------------------ | --------- |
> > > > | Beam Search                    | 11.95     |
> > > >
> > > > Also, note that **when complete programs need to be generated from the same prefix, it is only counted as one generated program because of caching**. SMCG-TD is a sampling-based algorithm and it could have a lot of identical partial programs in its population. Generating complete programs from identical partial programs would not contribute to the total number of generated programs.  Increasing its population size would indeed help increase the diversity of its population. However, it does not help much in generating more unique programs, as the number of generated programs only slightly increases when we increase the population size from 10 to 100 (Fig. 4 (left)).
> > > > PG-TD, on the other hand, would explicitly search for different unique problems by following different paths in the tree, which makes it have generally higher number of generated programs than SMCG-TD.
> > > >
> > > > It may also help understand the performance of the algorithms from their computational cost perspective. As shown in Fig. 4 (right), SMCG-TD consumes similar computation as PG-TD, while it does not generate as many unique samples and has a lower pass rate than PG-TD. **Given the same computational budget, PG-TD constantly generates the best programs.**
> > > >
> > > > We hope that the provided new experiments and additional explanations have convinced you of the merits of our work. Please do not hesitate to contact us if you have other concerns.
> > > >
> > > > We appreciate your time! Thank you so much!
> > > >
> > > > Best, Authors

---

> > > > > ### Comment · Reviewer_kmKW · 2022-11-23
> > > > > **Question regarding budget**
> > > > >
> > > > > Thanks for providing these additional results. I'm confused as to why SMCG-TD has exactly the same pass rate at all budgets. Is that a mistake? I would expect that adding a budget would reduce performance or at least change it in some way, but it looks like the algorithm is performing exactly the same under all budgets.
> > > > >
> > > > > For that matter, how many samples does SMCG-TD generate? Is it less than 512 (as suggested by Figure 4)? If so, limiting the budget to 512 or 1024 wouldn't actually be a fair comparison, since one of the approaches is using up the budget and the other is not. If SMCG-TD isn't generating enough programs, it seems like the right thing to do would be to increase the population size to make sure it generates at least 1024 programs, so that the budget is actually having some effect. (I think the ideal comparison would be to increase the population until it generates a similar number of programs as PG-TD with rollout=9, b=5.)

---

> > > > > > ### Author Response · Authors · 2022-11-25
> > > > > > **Response to Reviewer kmKW**
> > > > > >
> > > > > > We thank the reviewer for the follow-up questions. The pass rates of SMCG-TD (population=50) were **not** a mistake. The reviewer is correct that the number of generated programs for SMCG-TD (population=50) is generally lower than 512. So constraining the number of generated programs would not affect SMCG-TD especially when the population size is small.
> > > > > >
> > > > > > To address your concern, we have added results with smaller program generation budgets and also run SMCG-TD experiments with a larger population size of 200. The updated results are in Table F (updated) below. We have the same conclusion that the SMCG-TD has a better performance under small program generation budgets, while PG-TD achieves a higher pass rate when the program generation budget is 1024 and when the budget is not imposed.
> > > > > >
> > > > > > Table F (updated). Pass rates (%) of PG-TD and other algorithms under different program generation budgets, evaluated on the APPS introductory dataset.
> > > > > > | Algorithm \ Program Generation Budget | 128 | 256 | 512       | 1024      | $\infty$  |
> > > > > > | ------------------------ | ----- | ----- | ----- | ----- | ----- |
> > > > > > | PG-TD (rollout=3, b=1) | 22.53 | 23.11 | 23.16 | 23.18 | 23.16 |
> > > > > > | PG-TD (rollout=3, b=3) | 21.25 | 23.08 | 24.16 | 24.45 | 26.53 |
> > > > > > | PG-TD (rollout=6, b=3) | 21.23 | 22.93 | 24.42 | 24.73 | 26.93 |
> > > > > > | PG-TD (rollout=9, b=3) | 21.21 | 23.16 | 24.6  | **25.92** | **27.8**  |
> > > > > > | PG-TD (rollout=9, b=5) | 19.21 | 20.84 | 22.85 | 24.24 | 27.68* |
> > > > > > | SMCG-TD (pop=50)         | 23.44 | 23.81 | 23.76* | 23.76 | 23.76 |
> > > > > > | SMCG-TD (pop=100)        | **23.98** | 24.68 | 25.02 | 25.04 | 25.04 |
> > > > > > | SMCG-TD (pop=200)        | 23.95 | **25.17** | **25.32** | 25.35 | 25.4  |
> > > > > > | Sampling + Filtering     | 16.05 | 16.35 | 16.75 | -      | -      |
> > > > > >
> > > > > > | Algorithm \ Program Generation Budget | 5|
> > > > > > | ------------------------------ | --------- |
> > > > > > | Beam Search                    | 11.95     |
> > > > > >
> > > > > > (Note on Table F (updated): We find some pass rates decrease (marked with \*) when we increase the program generation budget or the beam size for PG-TD. This is caused by a slight overfitting issue. For example, for SMCG-TD (pop=50), when we increase the program generation budget from 256 to 512, the pass rate on the *public test cases* increases from 53.51% to 53.87%, while the pass rate on the *private test cases* drops from 23.81% to 23.76%.)
> > > > > >
> > > > > > We also show the average number of generated programs and the computation time of PG-TD and SMCG-TD in the table below. Since SMCG-TD is a sampling-based algorithm, with the same computational budget, it generates a smaller number of (unique) generated programs than PG-TD. Generating more unique programs (for example, reaching 1024) may require a significantly larger population size, which would make it even more computationally expensive.
> > > > > >
> > > > > >
> > > > > > Table G. The number of generated programs and the computation time of SMCG-TD, evaluated on the APPS introductory dataset.
> > > > > > |                   | # of Generated Programs | Computation Time (sec.) |
> > > > > > | ----------------- | ----------------------- | --- |
> > > > > > | PG-TD (rollout=3, b=1) | 260.67  | 1092.56 |
> > > > > > | PG-TD (rollout=3, b=3) | 763.05  | 1350.54 |
> > > > > > | PG-TD (rollout=6, b=3) | 1164    | 1995.61 |
> > > > > > | PG-TD (rollout=9, b=3) | 1895.33 | 3293.45 |
> > > > > > | PG-TD (rollout=9, b=5) | 3021.11 | 3820.25 |
> > > > > > | SMCG-TD (pop=50)  | 208.6                   | 1583.55    |
> > > > > > | SMCG-TD (pop=100) | 327.4                   | 2666.26    |
> > > > > > | SMCG-TD (pop=200) | 469.58                  | 3896.94    |
> > > > > >
> > > > > > (Note on Table G: We find the results in the paper did not count the computation times of runs which did not finish during the time allocation, but output some generated programs. We re-compute the computation times and the correct computation times are actually longer than the results in Fig. 4 (right).)
> > > > > >
> > > > > > We hope our response has addressed your concerns. We are happy to provide more results or clarifications if you have any additional comments or concerns. Thank you!
> > > > > >
> > > > > > Best,
> > > > > >
> > > > > > Authors

---

> > > > > > > ### Comment · Reviewer_kmKW · 2022-12-01
> > > > > > > **Discussion**
> > > > > > >
> > > > > > > Thanks for providing these additional results.
> > > > > > >
> > > > > > > Am I right in assuming that the program generation budget in Table F counts all programs, not just unique programs? That seems OK to me, since it does seem like an advantage of your approach that you can avoid duplicates. But if so, it would be good to distinguish between the type of count in table F and table G if the first is counting all programs and the second is counting unique programs. (It might be useful to report the total number of programs in table G as well, including duplicates, to give more context about how much repetition there is.)
> > > > > > >
> > > > > > > For the next version of the paper, I would suggest marking the numbers in Table F for which the budget was not reached (because the baseline didn't generate enough programs), perhaps with an asterisk or dagger or something and a clarification in the caption, since those are a bit misleading as written.
> > > > > > >
> > > > > > > In any case, I have updated my review to 8 with the understanding that these updated results will be included in the next version of the paper. It does seem like the proposed approach is an improvement over SMCG-TD when generating a large number of programs (both in terms of accuracy and computation time), and it is much better than sampling and filtering even for small numbers of programs.

---

> > > > > > > > ### Author Response · Authors · 2022-12-03
> > > > > > > > **Response to Reviewer kmKW**
> > > > > > > >
> > > > > > > > We thank the reviewer for reviewing our new results and the new comments. We will make Table F clear on how much program generation budget is actually used by each algorithm. We can convert the results to a figure like Fig. 4 (left) using the actual number of generation programs as the horizontal axis.
> > > > > > > >
> > > > > > > > We'd like to clarify that we count the number of **unique** generated programs in both Table F and G. In Table F, we stop the algorithm when the unique generated programs reach a budget. For a duplicate generation request, the algorithm would use the cached sequence without calling the Transformer generation function. We can also report another table counting duplicate function calls. In this case, for SMCG-TD, the number of total programs (counting duplicate ones) is roughly the size of the population * the length of the generated program ("roughly" because some programs in the population may finish earlier than others).
> > > > > > > >
> > > > > > > > We would like to thank the reviewer again for your time in reviewing our paper. Let us know if there is anything else we can provide or clarify.
> > > > > > > >
> > > > > > > > Best,
> > > > > > > >
> > > > > > > > Authors

---

### Official Review · Reviewer_Xzif · 2022-10-25

**Confidence:** 3
**Correctness:** 4
**Technical Novelty And Significance:** 4
**Empirical Novelty And Significance:** Not applicable
**Recommendation:** 8

**Clarity, Quality, Novelty And Reproducibility:**

The paper is largely clearly written and easy to follow. I found App. C.1 was very valuable to better understand some of the choices made and would urge the authors to include more of it in the main text of the paper.

As far as I am aware, this is a novel contribution (though I'm not familiar with the planning literature); the experiments are sound and support the claims made in the paper well. (I'd have appreciated more details on the timings of different models though).

**Strength And Weaknesses:**

* (+) This is an interesting direction, focusing on a core part of neural code generation that often receives little attention.
* (+) The empirical results indicate a significant improvement over baselines for the evaluated models.
* (~) I found the discussion of related work unsatisfactory - it's presented as a list of related papers, with little discussion of commonalities and differences to the presented work

**Summary Of The Paper:**

A new decoding strategy for language models for code is presented, based on the intuition of selecting tokens that lookahead search deems to be more likely to lead to correct sequence outputs. Additional ideas on managing the performance overhead of this strategy are presented. Finally, experimental results indicate that this new decoding strategy can outperform baselines by some margin.

**Summary Of The Review:**

Novel decoding algorithm, good experimental results, no major concerns: this paper should be accepted, as it will be of substantial interest to the ICLR subcommunity of researchers interested in code generation.

---

> ### Author Response · Authors · 2022-11-16
> **Response to Reviewer Xzif**
>
> Dear Reviewer Xzif,
>
> We thank the reviewer for the comments and insightful suggestions.
>
> **Q1. Related work.**
> > I found the discussion of related work unsatisfactory - it's presented as a list of related papers, with little discussion of commonalities and differences to the presented work.
>
> Thanks for your suggestions on related work. We have provided more details and highlighted the differences with blue colors in the revised version. We have reorganized the related work and emphasized the connections between our work and prior work.
>
> **Q2. More description of the MCTS algorithm in the main paper.**
> > I found App. C.1 was very valuable to better understand some of the choices made and would urge the authors to include more of it in the main text of the paper.
>
> We appreciate the reviewer's suggestion on the organization of the paper. We have summarized the MCTS algorithm in the main paper so that readers do not need to refer to the appendix to understand the basics of the algorithm.
>
> **Q3. Timing on different models.**
> > I'd have appreciated more details on the timings of different models though.
>
> Thanks for your suggestion on the timings of different models. The computation times of all the algorithms are reported in the right figure in Fig. 4. With the same computational budget, PG-TD generates better codes than the baseline algorithms.

---

### Official Review · Reviewer_fkta · 2022-11-01

**Confidence:** 4
**Correctness:** 3
**Technical Novelty And Significance:** 2
**Empirical Novelty And Significance:** 2
**Recommendation:** 3

**Clarity, Quality, Novelty And Reproducibility:**

Clarity: good
Quality: unclear based on results
Novelty: reasonable to my knowledge
Reproducibility: good

**Strength And Weaknesses:**

Strengths:
* The paper is clearly written and described.
* The website has a useful visualization and providing the code is appreciated (though there is a lot of code provided and maybe a subset would be preferred).
* The paper approaches an important research area, improving generation for LLMs and proposes a reasonable approach for it that does not require any additional training and minimal code complexity.
* The results demonstrate it outperforms baselines in performance.
* The results showing varied rewards is interesting and useful and a nice benefit of the approach.

Weaknesses
* The results are all performed with GPT-2 and seem to be a relatively weak LLM for coding. For coding in particular, where any small mistake may be catastrophic (in that it doesn't compile or run), I am unable to fully contextualize the benefits with a weak model (and one not trained on code). Codex for example would be a much stronger approach a minimize issues like the second sentence in the abstract “ Although the programs they generate achieve high token-matching-based scores, they often fail to compile or generate incorrect outputs.” Indeed, I ran two of the proposed tasks (Figure 1) and the circle problem on the website and *zero shot* Codex solved both. Furthermore it solved both with much more reasonable code than the PG-TD code and did it in under 5 seconds.
* The authors should report cost of running the algorithm.
* The overall success rate, particularly based on the examples shown in the paper, seems quite low for all methods.
* I believe all the results in this paper are zero-shot (or else I missed this), but presumably much of the issues would be solved with simple prompting with a few examples. The authors should compare to this.
* The examples of code shown are odd, in that it tends to generate unnecessarily complicated code for quite simple problems, stating `print(2*(r**2)-r**2)` instead of `print(r**2)`.
* Please discuss the failure modes, for both PG-TD and baselines, are they not compiling or failing test cases?
* I find pass rate an odd metric. These problems are simple and the input is constrained, so if the code doesn't solve them 100% then is it not incorrect code? If nothing else, the authors should report the number of tasks solved 100%.
* The computation times are quite slow and somewhat prohibitive, perhaps an improved LLM would help with this or a smaller one to search more quickly. Can you early terminate? What if a beam achieves 100% pass rate during search, do you stop?
* The caching I do not see as a major contribution. Writing a dictionary for cache[query or partial] should suffice unless I misunderstood.
* Though mostly clearly written, the statement of contributions, introduction, and abstract can be more clear what actually is done. What planning algorithm, what computationally efficient design mechanisms? These should be presented early and clearly for the reader.
* The method itself is reasonable interesting, integrating MCTS to the search, and it might be useful to show the approach on more varied problems, perhaps other coding challenges or BigBench benchmarks.


**Summary Of The Paper:**

This work proposes PG-TD, which integrates Transformer with MCTS for code generation. PG-TD works by generating a tree of possible programs, first generating high likelihood sequences and then evaluating them on test problems. It then selects the best and searches for more sequences until it completes the program. The authors also propose to cache completed queries as many are repeated during search. The authors demonstrate results of pass rate and computation time on an established coding challenge, demonstrating PG-TD outperforms GPT-3 baselines.


**Summary Of The Review:**

The authors propose a novel approach that is interesting and potentially useful, and the paper is clearly written. However, they only compare and leverage GPT-2 for coding problems, which leads to low performance across the board and making it unclear if they are solving a real problem -- they should run the examples with Codex and also with prompting. The algorithm is also quite slow, the code produced is overly complicated, and the metric of pass rate only is lacking (they should also report whether the code was correct). I do not believe caching to be a major contribution, but is presented as such and takes up a page. The paper may be improved by further demonstrating performance on other problems (perhaps not code) that have a reward structure.

---

> ### Author Response · Authors · 2022-11-16
> **Response to Reviewer fkta**
>
> Dear Reviewer fkta,
>
> We appreciate the reviewer for the detailed comments and insightful suggestions.
>
> **Q1: The Codex baseline.**
> > Codex for example would be a much stronger approach a minimize issues like the second sentence in the abstract “ Although the programs they generate achieve high token-matching-based scores, they often fail to compile or generate incorrect outputs.”
>
> We agree with the reviewer that Codex indeed generates better codes than the models used in our paper. The main reason that we do not use it for experiments is that it is not open-sourced and OpenAI limits the model's API usage. We have run both beam search and PG-TD algorithms using **Codex** as the backbone. We follow the "best practices" provided in the Codex guide to put the problem description into a block comment in the beginning, and ask Codex to complete the rest. As we expect, PG-TD helps Codex generate better codes. This further validates our claim that our method is model-agnostic and can help a pre-trained code generation Transformer generate better codes.
>
> Our results are reported in the following table. Due to the limits of the OpenAI APIs, we are only able to test the algorithms on a small set of data.
>
> Table A. Comparison between using different Transformers as backbones for code generation, evaluated on the first 42 APPS introductory problems.
>
> | Model       | Decoding Algorithm | Pass Rate | Strict Accuracy |
> | ------------ | ---------- | --------- | --------------- |
> | APPS GPT-2 (1.5B) | Beam Search | 9.38          | 0.0                |
> | APPS GPT-2 (1.5B) | PG-TD | 30.99          | 2.38                |
> | Codex (code-davinci-002) | Beam Search | 34.56        | 9.52                |
> | Codex (code-davinci-002) | PG-TD | 47.17         | 23.80                |
>
>
>
> **Q2. Computational Analysis.**
> > The authors should report cost of running the algorithm.
>
> Thanks for your advice. Indeed, we have reported the computational analysis in Fig. 4. In the right of Fig. 4, the computation times of the variations of PG-TD and the baseline algorithms are reported. With the same computational budget, PG-TD generates better codes than the baseline algorithms.
>
>
> **Q3. About zero-shot setting and comparison with prompting baselines.**
> > I believe all the results in this paper are zero-shot (or else I missed this), but presumably much of the issues would be solved with simple prompting with a few examples. The authors should compare to this.
>
> The large language models used in our PG-TD is **not** zero-shot. Since the large language models (GPT-1.5B and GPT-Neo) were first pre-trained on massive code data and then finetuned on APPS training datasets to generate solutions to the coding problems. Note that such a setting was also used in the APPS paper [1] and AlphaCode [2]. It is a good idea to add prompting baselines for the general language models. However, **It has been reported in Table 2 of the original APPS paper [1] that a standard few-shot prompting method with GPT-3 (175B) achieves much lower pass rates than finetuning (0.55 vs 10.15).** This also confirms the necessity to use the proposed PG-TD with a *finetuned* transformer to achieve good performance.
>
> [1]. Dan Hendrycks, Steven Basart, Saurav Kadavath, Mantas Mazeika, Akul Arora, Ethan Guo, Collin Burns, Samir Puranik, Horace He, Dawn Song, and Jacob Steinhardt. Measuring coding challenge competence with APPS. NeurIPS, 2021.
>
> [2]. Yujia Li, David Choi, Junyoung Chung, Nate Kushman, Julian Schrittwieser, R ́emi Leblond, Tom Eccles, James Keeling, Felix Gimeno, and Agustin Dal Lago. Competition-Level Code Generation with AlphaCode. arXiv preprint arXiv:2203.07814, 2022.
>
>
> **Q4: About failure modes.**
> > Please discuss the failure modes, for both PG-TD and baselines, are they not compiling or failing test cases?
>
> In the table below, we report the average percentages of test cases where compilation errors and runtime errors occur. We use the error analysis utility provided in the original APPS codebase. As we expect, PG-TD executes the generated programs and finds the ones with higher pass rates, so it dramatically reduces the percentage of both types of errors.
>
>
> Table C. Failure case analysis. The percentages are averaged over the APPS introductory dataset.
> |                                | Compilation Error (%) | Runtime Error (%) |
> | ------------------------------ | --------------------- | ----------------- |
> | APPS GPT-2 (1.5B), Beam Search | 5.58                  | 32.95             |
> | APPS GPT-2 (1.5B), Sampling + Filtering    | 4.68                  | 27.38                  |
> | APPS GPT-2 (1.5B), PG-TD       | 1.93                  | 19.50             |

---

> > ### Author Response · Authors · 2022-11-16
> > **Response to Reviewer fkta (continued)**
> >
> > **Q5: On evaluation metrics.**
> > > I find pass rate an odd metric. These problems are simple and the input is constrained, so if the code doesn't solve them 100% then is it not incorrect code? If nothing else, the authors should report the number of tasks solved 100%.
> >
> > Thanks for your suggestion on reporting percentages of problems in which code-generation algorithms pass all test cases. Indeed, this is the **strict accuracy** metric we used in the experiment sections (Table 1). We use pass rates as rewards as they provide more dense signals than strict accuracies. However, we do find that the generated programs with higher pass rates also tend to achieve higher strict accuracies.
> >
> >
> > **Q6: About termination conditions.**
> > > The computation times are quite slow and somewhat prohibitive, perhaps an improved LLM would help with this or a smaller one to search more quickly. Can you early terminate? What if a beam achieves 100% pass rate during search, do you stop?
> >
> > We agree that early terminating when finding a solution with a pass rate of 1 would certainly make our algorithms more efficient. In the table below, we find enabling early termination does improve the algorithms' computational efficiencies. We will include this implementation choice in the paper. We expect early termination would be more helpful when the code generation algorithms can solve most of the problems (achieving pass rates of 1 on most of them), which is not the case for the current model.
> >
> > Table D. Computation times with and without early stopping, using APPS GPT-2 (1.5B).
> > |                                                  | Computation Time (sec.) |
> > | ------------------------------------------------ | ----------------------- |
> > | PG-TD without early termination (as in paper)    | 1177.99                        |
> > | PG-TD with early termination                     | 986.37                        |
> > | Sampling without early termination (as in paper) | 2154.23                        |
> > | Sampling with early termination                  | 2081.25                        |
> >
> > On the other hand, for n@k metrics (reported in Table 6), the code-generation agent may not want to terminate early. Under these metrics, the agent can submit multiple solutions and be considered as solving the problem if any of the submitted solutions passes all the **private** test cases. The agent can benefit from finding multiple solutions that pass all the **public** test cases and submit all of them for evaluation.
> >
> > **Q7. Caching algorithms.**
> > > The caching I do not see as a major contribution. Writing a dictionary for cache[query or partial] should suìce unless I misunderstood.
> >
> >
> > We find that **simply caching all the generated sequences and reusing them later would not be sufficient**. In *sequence caching*, we do cache all the generated sequences. These sequences can be reused in the evaluation step in MCTS when we need to generate complete programs from partial programs. Importantly, we justified that reusing the cached sequences are greedily-optimal when the beam size is 1, and the cached sequences may not be greedily-optimal (that is, the cached sequences can be different from the sequences generated by rerunning the beam search algorithm) when the beam size is larger than 1. We have more detailed discussions in Sec. C.2.
> >
> > We also contributed to designing *tree structure caching* by identifying the tree structure shared by the Transformer and MCTS, so that the most-likely next tokens of partial programs found by the beam search algorithm can be shared with MCTS for building the tree later. This method is unique to combining MCTS with beam search, and to the best of our knowledge, how this combination can be made more efficient is not addressed in the literature.

---

> > ### Author Response · Authors · 2022-11-23
> > **Looking forward to more discussion**
> >
> > Dear Reviewer fkta,
> >
> > We would like to thank you again for your feedback and questions. We have responded to your comments a few days ago. We were wondering whether these results and clarifications have addressed your questions and concerns. Please do not hesitate to let us know if there is anything else we can provide and clarify. Thank you!

---

> > ### Author Response · Authors · 2022-12-08
> > **Looking forward to more discussion**
> >
> > Dear reviewer fkta,
> >
> > Thank you again for your reviews on the paper. In our response above, we have responded to your comments by providing the following results and clarifications.
> > * We have added new experiments using the CodeX model as the backbone transformer;
> > * We have provided other experimental results including failure mode analysis and effects of using early termination;
> > * We have provided more explanations on computational complexity, evaluation metrics, and the caching mechanisms.
> >
> > We were wondering whether these results and clarifications have addressed your questions and concerns. Please do not hesitate to let us know if there is anything else we can provide. Thank you!
> >
> > Authors

---

### Author Response · Authors · 2022-11-16
**General Response to All Reviewers: Results and Revisions**

In this discussion period, we have the following new results and revisions to the paper.
* **Results on Codex** (Reviewer fkta, FcyD). The reviewers have raised the question that our paper does not use the most capable code-generation Transformer (for example, OpenAI Codex). The main reason is that Codex is **not open-sourced** and OpenAI limits the frequency of calling their code generation APIs (20 times per second). So we cannot conduct large-scale experiments using the Codex model. In this rebuttal, we strive to perform more experiments using Codex as the backbone code-generation Transformer and generating codes using beam search vs. PG-TD. We are able to perform the comparison on a subset of APPS problems and are still able to find that **using Codex, PG-TD generates much higher-quality codes compared with using beam search**. This result further confirms that our algorithm is model-agnostic.

Table A. Comparisons between using different Transformers as backbones for code generation, evaluated on the first 42 APPS introductory problems.

| Model       | Decoding Algorithm | Pass Rate | Strict Accuracy |
| ------------ | ---------- | --------- | --------------- |
| APPS GPT-2 (1.5B) | Beam Search | 9.38          | 0.0                |
| APPS GPT-2 (1.5B) | PG-TD | 30.99          | 2.38                |
| Codex (code-davinci-002) | Beam Search | 34.56        | 9.52                |
| Codex (code-davinci-002) | PG-TD | 47.17         | 23.80                |

* **Sampling-based MCTS** (Reviewer kmKW). In the evaluation step of our MCTS algorithm, we generate greedily-optimal solutions from a partial program using beam search. We could instead use a sampling approach, which would be a more faithful implementation of MCTS as suggested by the reviewer. We experimented with using sampling in the generation step and found that **using beam search in the evaluation step of the tree search algorithm does have a better performance than using sampling**.

Table B. PG-TD using beam search vs. sampling in the evaluation step, evaluated on the first 500 APPS introductory problems.
| Decoding Algorithm     | Pass Rate | Strict Accuracy |
| ---------------------- | --------- | --------------- |
| PG-TD using Beam Search (as in paper) | 32.757          | 13.20                |
| PG-TD using Sampling | 15.69          | 5.0                |

* **Consistent baseline hyper-parameters** (Reviewer kmKW). In the experiments in our original submission, we use top-k sampling where k=5, hoping this benefits sampling. For other experiments, we use k=3. We now set k=3 for both our algorithms and the baseline algorithms, that is, only the first three most-likely tokens are considered in each generation step (except when we explicitly change the tree width in the variations of PG-TD in Fig. 4). We have updated Fig. 4 to reflect the change.
* **Algorithm failure mode analysis** (Reviewr fkta). We have re-evaluated the generation solutions and reported the types of errors when the generated programs fail to run. As we expect, PG-TD has a lower error rate compared with other baseline algorithms in terms of both compilation error and runtime error.

Table C. Failure case analysis. The percentages are averaged over the APPS introductory dataset.
|                                | Compilation Error (%) | Runtime Error (%) |
| ------------------------------ | --------------------- | ----------------- |
| APPS GPT-2 (1.5B), Beam Search | 5.58                  | 32.95             |
| APPS GPT-2 (1.5B), Sampling + Filtering    | 4.68                  | 27.38                  |
| APPS GPT-2 (1.5B), PG-TD       | 1.93                  | 19.50             |
* **More description of the MCTS algorithm in the main paper** (Reviewer Xzif, FcyD). We have summarized the description of the MCTS algorithm in the main paper so that readers do not need to refer to the appendix to understand the basics of the algorithm.
* **Qualitative analysis** (Reviewer fkta, FcyD). The examples we show in the paper are indeed easy problems where APPS GPT-2 Transformers fail. We added a new example where **the Codex model fails to solve the problem using beam search, but successfully solves the problem (pass rate = 1) using PG-TD**. We include the example in Fig. 15 in the revision.

Please see our detailed responses to all the reviewers below.

---

### Author Response · Authors · 2022-11-16
**General Response to All Reviewers**

We thank all the reviewers for taking time to review our paper and providing insightful feedback. We are glad that reviewers recognized the following contributions.
* **Task**. The task we consider in this paper is important and has a broad interest.

> The paper approaches an important research area. (Reviewer fkta)

> This is an interesting direction, focusing on a core part of neural code generation that often receives little attention. (Reviewer Xzif)

> The problem is important and has a broad interest. (Reviewer FcyD)

* **Method**. Our approach of integrating LLM with MCTS for code generation is novel and reasonable.

> The idea of treating code generation as an RL problem, where input-output pairs provide the reward, is novel and related to much work in program synthesis. (Reviewer FcyD)

> Integrating LM generation with MCTS is a good idea, allowing the search process to focus on prefixes with a higher chance of producing good programs. (Reviewer kmKW)

> The paper approaches an important research area, improving generation for LLMs and proposes a reasonable approach for it that does not require any additional training and minimal code complexity. (Reviewer fkta)

* **Experiment**. We show empirical evidence that our method significantly outperforms baseline algorithms.

> The results demonstrate it outperforms baselines in performance. (Reviewer fkta)

> The empirical results indicate a significant improvement over baselines for the evaluated models. (Reviewer Xzif)

> Their method seems to slightly outperform baseline methods given a fixed runtime budget, without needing to retrain the transformer model. (Reviewer kmKW)


**Novelty.**
First, we **never claim** the application of MCTS in neural program synthesis is our main contribution (as discussed in the related work section). Instead, our work aims to answer the question **if MCTS can be integrated with large language models (models at the scale of ~2B parameters, and Codex in our new results with ~12B parameters) for competitive-level code generation, and if the Transformer + MCTS framework outperforms the well-accepted Sampling + Filtering scheme that has shown successes in the literature (used by AlphaCode, etc.).** To this end, we propose concrete designs of the framework to achieve strong performance (how the information provided by the Transformer should be used by MCTS, and using beam search in the evaluation step instead of using the standard sampling method, etc.), as well as improve the efficiency of the overall framework (the caching methods).

We also want to emphasize that our proposed algorithm, PG-TD, is **model-agnostic**. The goal is not to use an extremely large model (with sufficient training data) to achieve state-of-out-art performance in code generation. Our work focuses on the following problem. **Given a pre-trained code-generation model (which could have a small size or does not perform perfectly on a dataset), can we improve its performance by only changing the decoding method with the help of some test cases**, without necessarily increasing the size of the model, finetuning the model, and using more human-written sample solutions as training data?

We emphasize our contributions as follows.
1. We are the first to combine MCTS with large language models for general-purpose programming language generation from natural language descriptions. Our algorithm achieves a significantly better performance than the existing well-accepted baselines. Reviewer kmKW has recognized the empirical value of showing that MCTS works well with LLM for code generation.
2. We have designed the interfaces between the MCTS and LLM and dealt with the unique challenges of making the framework computationally efficient.
3. We demonstrated the versatility of PG-TD by optimizing other coding targets like generating shorter codes and codes with more comments (Fig. 5 in paper), and also showed that codes generated by PG-TD can help further finetune the code-generation Transformer, achieving a self-learning framework without requiring extra human-written training samples (Table 4 in paper).

---

### Decision · Program_Chairs · 2023-01-20

**Decision:**

Accept: poster

**Justification For Why Not Higher Score:**

The methodology in which this paper is working was judged to be of limited interest.

**Justification For Why Not Lower Score:**

The methodology does have an established published tradition and the paper is unlikely to benefit from revision.

**Metareview: Summary, Strengths And Weaknesses:**

Strengths: This is an interesting direction, focusing on a core part of neural code generation that often receives little attention.
The empirical results indicate a significant improvement over baselines for the evaluated models.  The idea of treating code generation as an RL problem, where input-output pairs provide the reward, is novel and relevant to much work in program synthesis. The authors response was strong and addressed many of the weaknesses.

Weaknesses: The main weakness was seen to be the use of both an English description of the task to be solved as well as a set of unit tests (input/output pairs).  Writing code with both a natural language language description and provided unit tests seems a poor match to actual coding, or at least a less challenging problem that working from English alone.

In the end it was decided that this paper is working in an established paradigm and within that paradigm it is a fairly complete paper and would not benefit greatly from another revision.  One reviewer would still like to see a demonstration of generating unit tests from natural language descriptions as was claimed to be possible in an author response.

**Note From Pc:**

if the above contains the word "oral" or "spotlight" please see: "oral" presentation means -> notable-top-5% and "spotlight" means -> notable-top-25%. As stated in our emails, we are disassociating presentation type from AC recommendations

**Summary Of Ac-Reviewer Meeting:**

At the outset of the meeting we had ratings of 3,3,8,8.  One reviewer, Reviewer fkta, who gave the paper a 3, had not been responsive to author's rebuttal and was not responsive to the request for a meeting.  The other rating of 3 was from Reviewer FcyD.  Reviewer FcyD, and Reviewer Xzif both work in closely related areas but had opposite ratings.  After the discussion of pros and cons, as described above, Reviewer FcyD agreed that the paper was unlikely to benefit from revision and was within existing (but questionable) methodology.  He therefore agreed to improve his score to a 6.  The only reviewer absent from the meeting was Reviewer fkta.